# A late-stage assembly checkpoint of the human mitochondrial ribosome large subunit

Pedro Rebelo-Guiomar [1], Simone Pellegrino [2,3,4], Kyle C. Dent [2,3,4,8], Aldema Sas-Chen [5,9], Leonor Miller-Fleming [1], Caterina Garone [1,10], Lindsey Van Haute [1], Jack F. Rogan [6], Adam Dinan [7], Andrew E. Firth [7], Byron Andrews [6], Alexander J. Whitworth [1], Schraga Schwartz [5], Alan J. Warren [2,3,4] & Michal Minczuk [1✉]

Many cellular processes, including ribosome biogenesis, are regulated through post-transcriptional RNA modifications. Here, a genome-wide analysis of the human mitochondrial transcriptome shows that 2′-O-methylation is limited to residues of the mitoribosomal large subunit (mtLSU) 16S mt-rRNA, introduced by MRM1, MRM2 and MRM3, with the modifications installed by the latter two proteins being interdependent. MRM2 controls mitochondrial respiration by regulating mitoribosome biogenesis. In its absence, mtLSU particles (visualized by cryo-EM at the resolution of 2.6 Å) present disordered RNA domains, partial occupancy of bL36m and bound MALSU1:L0R8F8:mtACP anti-association module, allowing five mtLSU biogenesis intermediates with different intersubunit interface configurations to be placed along the assembly pathway. However, mitoribosome biogenesis does not depend on the methyltransferase activity of MRM2. Disruption of the MRM2 *Drosophila melanogaster* orthologue leads to mitochondria-related developmental arrest. This work identifies a key checkpoint during mtLSU assembly, essential to maintain mitochondrial homeostasis.

[1] MRC Mitochondrial Biology Unit, University of Cambridge, Cambridge Biomedical Campus, Keith Peters Building, Hills Rd, Cambridge CB2 0XY, UK. [2] Cambridge Institute for Medical Research, University of Cambridge, Cambridge Biomedical Campus, Keith Peters Building, Hills Rd, Cambridge CB2 0XY, UK. [3] Wellcome Trust – MRC Stem Cell Institute, Cambridge Biomedical Campus, Jeffrey Cheah Biomedical Centre, Puddicombe Way, Cambridge CB2 0AW, UK. [4] Department of Haematology, School of Clinical Medicine, University of Cambridge, Cambridge Biomedical Campus, Jeffrey Cheah Biomedical Centre, Puddicombe Way, Cambridge CB2 0AW, UK. [5] Department of Molecular Genetics, Weizmann Institute of Science, Rehovot 76100, Israel. [6] STORM Therapeutics Limited, Babraham Research Campus, Moneta Building, Cambridge CB22 3AT, UK. [7] Department of Pathology, University of Cambridge, Tennis Court Road, Cambridge CB2 1QP, UK. [8] Present address: MRC Laboratory of Molecular Biology, Cambridge Biomedical Campus, Francis Crick Avenue, Cambridge CB2 0QH, UK. [9] Present address: Shmunis School of Biomedicine and Cancer Research, The George S. Wise Faculty of Life Sciences, Tel Aviv University, Tel Aviv 6997801, Israel. [10] Present address: Department of Medical and Surgical Sciences, University of Bologna, Bologna 40137, Italy. ✉email: michal.minczuk@mrc-mbu.cam.ac.uk

The human mitochondrial genome is encoded in multiple copies of ~16.6 kb circular double-stranded DNA molecules (mtDNA) present in mitochondrial nucleoids in the mitochondrial matrix. Expression of this genome entails several, highly regulated processes, with newly synthesised transcripts being cleaved, chemically modified, polyadenylated and further matured in neighbouring structures known as mitochondrial RNA granules (MRGs). The assembly of mitochondrial ribosomes also takes place in MRGs, with mitoribosomal proteins (MRPs) and biogenesis factors engaging in a complex process, which has not yet been fully characterised[1–3].

Similar to other systems, the mitochondrial ribosome is composed of a small (mtSSU) and a large (mtLSU) subunit, with their core rRNAs, 12S and 16S mitochondrial (mt-) rRNAs, respectively, surrounded by MRPs (30 for the mtSSU and 52 for the mtLSU). While the RNA components of the mitoribosome are mitochondrially-encoded, all MRPs and assembly factors are encoded in the nuclear genome, thus requiring coordination between two genomes for the assembly of these macromolecular complexes. The mammalian mitochondrial ribosome is endowed with a number of specific features. While RNA makes up most of the composition of bacterial and cytosolic eukaryotic ribosomes, mammalian mitochondrial ribosomes present a more elaborate protein shell, which aids coping with the oxidative microenvironment. Almost half of these MRPs are evolutionarily exclusive to mitochondrial ribosomes, some of which were repurposed and accreted during reductive genome evolution[4,5]. In addition, several mitochondrial ribosomal proteins that share homology to other translation systems present mitochondria-specific extensions, structurally compensating truncated rRNA segments[4–7]. The peptide exit tunnel is lined with hydrophobic residues, which stabilise the highly hydrophobic nascent mitochondrial peptides[4]. Furthermore, instead of 5S rRNA, structurally similar mtDNA-encoded tRNAs occupy an equivalent region in the central protuberance of the mtLSU. Depending on the organism and availability, mt-tRNA$^{Val}$ or mt-tRNA$^{Phe}$ are incorporated, most likely due to their genomic proximity to mt-rRNA genes and consequent near stoichiometric presence of their transcripts[4,8].

As other RNA classes, mt-rRNAs contain modified ribonucleotides which are post-transcriptionally introduced by a set of enzymes. Unlike the bacterial and eukaryotic cytoplasmic counterparts which rRNAs carry tens of post-transcriptional modifications, there are only 10 modified residues in mammalian mt-rRNAs, all clustering in functionally relevant regions of the mitoribosome, including the A- and P-loops of the peptidyl transferase centre (PTC) in the 16S mt-rRNA, or the decoding centre in the 12S mt-rRNA[9]. Methylation of the 2′-hydroxyl group of ribonucleotides (2′-O-methylation) has been detected in G2815, U3039 and G3040 of 16S mt-rRNA (human mtDNA numbering) and the involved enzymes identified as MRM1, MRM2 and MRM3, respectively[10–12]. However, the role of the modifications and the enzymes introducing them has remained elusive and lacking molecular and mechanistic insight. Furthermore, mutations in *MRM2* have been implicated in human pathogenesis, in an ever-growing group of patients affected by mitochondrial diseases, for which pathogenesis mechanisms are not yet completely understood with molecular detail[13].

With this, we set out to investigate the role of MRM2 in the biogenesis and function of human mitochondrial ribosomes. We show that MRM2, but not its methyltransferase activity, is essential for mtLSU biogenesis, as it remodels the conformation of 16S rRNA in the late stages of this process. In the absence of MRM2, a severe mitochondrial dysfunction phenotype with underlying defects in mitochondrial translation is observed, with the mtLSU being trapped in immature assembly states where

domains IV and V of 16S mt-rRNA are unstructured, and an anti-association module is present, preventing premature engagement of these immature particles in translation. Finally, using a *Drosophila melanogaster* model, we show that MRM2 is essential for the development and homeostasis of neuronal and muscular tissues, often affected in mitochondrial diseases.

## Results

**2′-O-Methylation in the human mitochondrial transcriptome.** To investigate the presence of nucleotides with 2′-O-methylated riboses in mitochondrial RNAs, we performed a transcriptome-wide profiling in cells lacking three known 2′-O-methyltransferases, MRM1, MRM2 and MRM3[10–12] (Fig. 1 and Supplementary Fig. 1). To this end, we used a combination of two high throughput sequencing-based methods. First, an adaptation of RiboMeth-Seq[14,15], which relies on the property of 2′-O-methylated riboses to resist alkaline hydrolysis; thus, these modified residues exhibit high 'cleavage protection' scores compared to non-methylated sites. Second, a variation of 2OMe-Seq[16], which exploits the reverse transcription (RT) stops elicited by the 2′-O-methylation when using limiting concentrations of deoxyribonucleotide triphosphates (dNTPs); in this method, a higher ratio of RT-stop in low- versus high-dNTP conditions correlates with a higher level of methylation. By combining these two methods, the presence of the three known 2′-O-methylated sites in the 16S mt-rRNA was confirmed (Fig. 1a). Our data also show that MRM1 is responsible for catalysing the ribose methylation of G2815, which is the only position affected by its ablation. Similarly, modification of G3040 is only affected in the absence of MRM3. However, knocking-out either *MRM2* or *MRM3* impacted on the presence of Um3039. While we observed a slight reduction of MRM1 and MRM2 protein steady-state levels in *MRM2* and *MRM3* knock-out cells, respectively (Supplementary Fig. 1), we note that (i) no reduction in the modification status of MRM1 target (G2815) is detected upon *MRM2* knock-out and (ii) the very strong reduction in modification of the MRM2 target (U3039) in *MRM3* knock-out cells is unlikely to be attributable to a modest downregulation of MRM2. This, however, suggests that U3039 methylation by MRM2 depends on prior modification of G3040 by MRM3. No additional MRM1-, MRM2- or MRM3-dependent 2′-O-methylation sites were detected across the mitochondrial transcriptome using these methods (Fig. 1b).

**Ablation of MRM2 impairs mitochondrial respiration.** Given the association of MRM2 in human mitochondrial pathology[13], we focused on understanding its role in mitochondrial function. We first evaluated the functional status of mitochondria in *MRM2* knock-out cells by supplying cells with glucose or galactose as their main carbon source[17]. When cultured in high glucose medium, these cells proliferated at a slower rate, and failed to survive in galactose medium, consistent with a profound and generalised mitochondrial dysfunction (Fig. 2a). This was corroborated by constitutively high extracellular acidification rate (ECAR, indicative of anaerobic respiration), which was insensitive to inhibition of mitochondrial respiration or uncoupling, and nominal oxygen consumption rate (OCR) (Fig. 2b). The detected impairment of cellular respiration was characterised in more detail by assessing the activity of mitochondrial respiratory chain (MRC) complexes (Fig. 2c). While the activity of complex II, the subunits of which are encoded exclusively in the nuclear genome, was not decreased upon depletion of MRM2, that of complexes I and III, which contain mtDNA-encoded core subunits, were significantly decreased (<30% relative to control). Enzymatic activity of complex IV was below detection level and thus it was

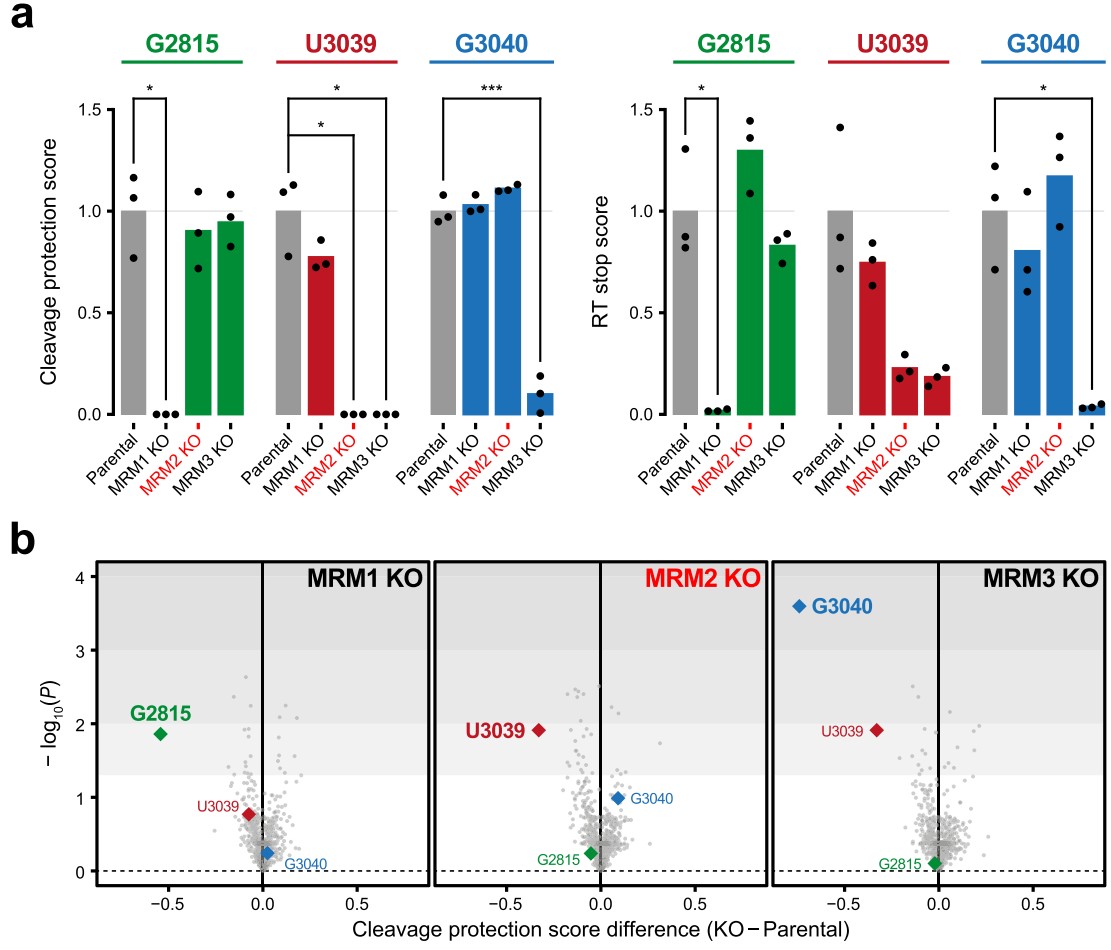

**Fig. 1 The human mitochondrial transcriptome contains three 2′-O-methylated residues, modified by MRM1, MRM2 and MRM3. a** Determination of ribose methylation levels of known targets of MRM1 (G2815), MRM2 (U3039) and MRM3 (G3040). Experimental values and mean are shown. Statistical significance was assessed using two-tailed Student's $t$ test (∗: $P ≤ 0.05$; ∗∗: $P ≤ 0.01$; ∗∗∗: $P ≤ 0.001$; Cleavage protection score: G2815 Parental vs *MRM1* KO $P = 0.0138$, G3039 Parental vs *MRM2* KO $P = 0.0122$, G3039 Parental vs *MRM3* KO $P = 0.0122$, G3040 Parental vs *MRM3* KO $P = 0.0003$; RT stop score: G2815 Parental vs *MRM1* KO $P = 0.0236$, G3040 Parental vs *MRM3* KO $P = 0.0235$). **b** Mitochondrial transcriptome-wide evaluation of 2′-O-methylation in cells devoid of MRM1, MRM2 or MRM3 ($n = 3$ for each cell line). $P$ values were determined by two-tailed Student's $t$ test. Source data are provided as a Source Data file.

not possible to accurately determine its value spectrophotometrically. Separation of cellular components in native conditions coupled to in-gel detection of cytochrome c oxidase activity showed that this enzymatic deficiency is due to the virtual absence of assembled complex IV (Supplementary Fig. 2a). In agreement with this, quantitative analysis of the mitochondrial proteome further revealed a reduction in the steady-state levels of MRC components (Fig. 3a), with the protein presenting the most severe observable down-regulation being MT-CO2, one of the mtDNA-encoded core subunits of complex IV. Taken together, these data show that MRM2 is indispensable for mitochondrial respiratory function in human cells.

**MRM2 is essential for mitochondrial translation**. The involvement of MRM2 in 16S mt-rRNA U3039 2′-O-methylation led us to hypothesise that the observed generalised oxidative phosphorylation (OxPhos) defects stem from the perturbation of mitochondrial ribosome-related mechanisms. To test this, the synthesis of mtDNA-encoded proteins was assessed by metabolic labelling of nascent peptides in *MRM2* knock-out and parental cells (Fig. 3b and Supplementary Fig. 2b). The near absence of *de novo* synthesised mitochondrial proteins observed upon MRM2 depletion revealed a considerable dysfunction of mitochondrial

translation. The degree to which each mtDNA-encoded protein was affected did not depend on its length (Supplementary Fig. 2c), suggesting that this perturbation is not related to translation elongation[18]. Interestingly, we observed a strong upregulation of steady-state levels of TOM22 in *MRM2* knock-out cells (Fig. 3c), consistent with an increase in mitochondrial mass in this sample, with the latter being supported by staining with a dye (MitoTracker Green) which accumulates in mitochondria irrespective of mitochondrial membrane potential (Supplementary Fig. 2d, e). The increase in mitochondrial mass could represent a compensatory effect resulting from strong inhibition of mitochondrial translation[19].

In order to determine which translation step is affected by the ablation of MRM2, the position of mitochondrial ribosomes along transcripts was determined using high-throughput mitochondrial ribosome footprinting—mitoRibo-Seq[18]. Consistent with the previous findings, the quantity of ribosomes engaged in translation was considerably reduced in the absence of MRM2, where the average mitoribosome occupancy on mt-mRNAs was 13.9% ± 7.6% (minimum: 1.6%, median: 12.0%, maximum: 35.5%) of that observed in parental samples (Fig. 3c and Supplementary Fig. 3). No bias in the occupancy of mitoribosomes towards the 5′ or 3′ regions of the coding sequences was

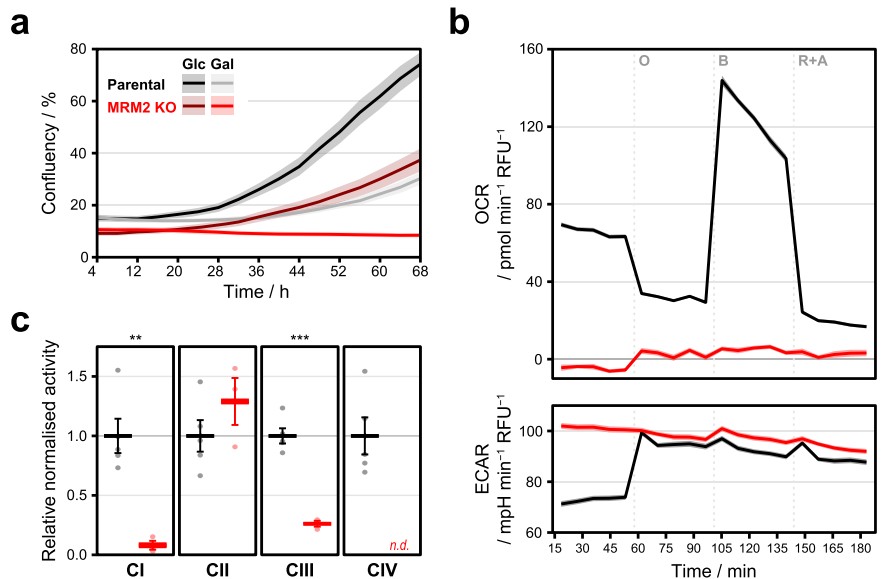

**Fig. 2 Ablation of MRM2 leads to a severe impairment of cellular respiration and dysfunction of MRC complexes. a** Cell proliferation in glucose (Glc) or galactose (Gal) media. Data are presented as mean ± SD ($n = 8$, independent measurements). **b** Oxygen consumption rate (OCR) and extracellular acidification rate (ECAR) measurements in the presence of mitochondrial respiration inhibitors (O: oligomycin; B: BAM15; R + A: rotenone and antimycin A). Data are presented as mean ± SD ($n = 22$ for parental, $n = 23$ for *MRM2* knock-out, independent measurements). **c** Spectrophotometric determination of the activity of MRC complexes (CI: complex I, NADH:ubiquinone oxidoreductase; CII: complex II, succinate:ubiquinone oxidoreductase; CIII: complex III, ubiquinol:cytochrome *c* oxidoreductase; CIV: complex IV, cytochrome *c* oxidase). n.d.: no data. Experimental values and mean ± SD are shown. Statistical significance was assessed using two-tailed Student's *t* test (∗∗: $P \leq 0.01$; ∗∗∗: $P \leq 0.001$; CI Parental vs MRM2 KO $P = 0.0023$, CIII Parental vs *MRM2* KO $P = 0.0001$). Data from parental and *MRM2* knock-out (KO) cell lines are represented in black and red, respectively. Source data are provided as a Source Data file.

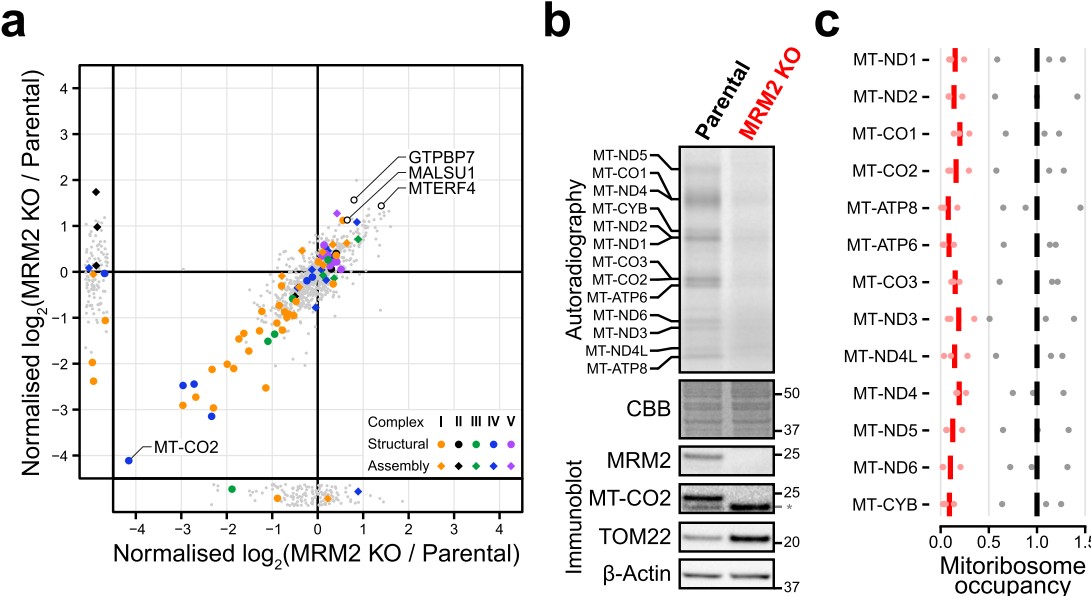

**Fig. 3 OxPhos complexes are structurally compromised in the absence of MRM2 due to impaired mitochondrial translation. a** Quantitative mass spectrometric analysis of the mitochondrial proteome of *MRM2* knock-out and parental control cells. Datapoints corresponding to structural components (circles) or assembly factors (diamonds) of OxPhos complexes, and other proteins of interest are highlighted. **b** Metabolic labelling of *de novo* translated mitochondrial proteins (top, quantification in Supplementary Fig. 2a, b) and immunoblotting assessment of steady-state levels of mitochondrial proteins (bottom). Molecular weights of protein standards are presented in kDa to the right of each blot; an asterisk marks a band generated in a previous blot for TOM22. Coomassie brilliant blue (CBB) staining is shown as a loading indicator. This experiment was replicated three times with similar results. **c** Occupancy of mitochondrial transcripts by mitochondrial ribosomes. Experimental values and mean are shown. Data from parental and *MRM2* knock-out (KO) cell lines are represented in black and red, respectively. Source data are provided as a Source Data file.

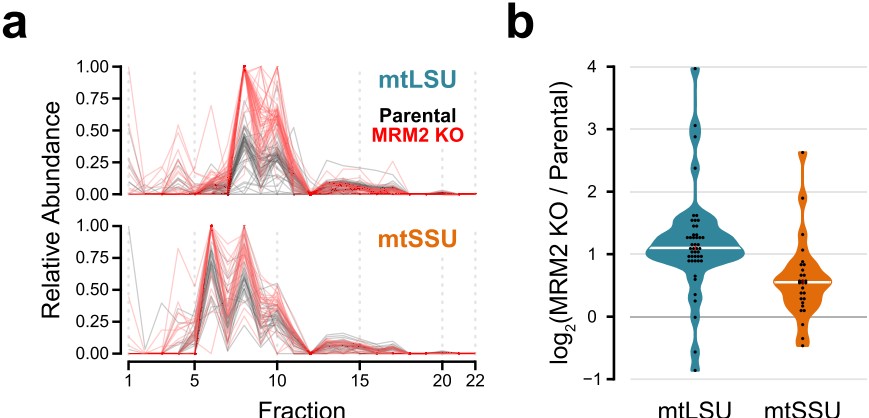

**Fig. 4 mtLSU accumulates as mature-like particles in the absence of MRM2. a** Quantitative gradient fractionation analysis by mass spectrometry (qDGMS) profile of mitoribosomal proteins. A heatmap of individual profiles is shown in Supplementary Fig. 4 and extended data is available in Supplementary Data 2. **b** Assessment of the enrichment of mitoribosomal proteins by subunit. The average from two reciprocal labelling experiments is presented. White horizontal bars represent the median of each distribution. Source data are provided as a Source Data file.

observed. Moreover, the pattern of ribosome occupancy along the body of mt-mRNAs was similar between *MRM2* knock-out and parental samples, with no codon-specific stalling. The metabolic labelling and mitoRibo-Seq experiments show that a profound impairment of mitochondrial translation underlies the mitochondrial dysfunction phenotype, possibly by perturbing the biogenesis of the mitochondrial ribosome.

**MRM2 is involved in the late stages of mtLSU assembly**. To investigate the mechanistic basis for the defect in mitochondrial translation, the composition of mitochondrial ribosomes was assessed using quantitative density gradient analysis by mass spectrometry – qDGMS[20]. While no shifts in the sedimentation profile of mitoribosomal components were observed, proteins of the mtLSU were enriched upon MRM2 depletion (Fig. 4, Supplementary Fig. 4 and Supplementary Data 2), indicating the accumulation of mature or near-mature forms of this subunit. Given the functional impairment of apparently structurally sound mtLSU particles, these were isolated from *MRM2* knock-out cells and their structure determined to 2.58 Å (Supplementary Fig. 5) using cryo-electron microscopy (cryoEM) single particle analysis. Owing to the preferential orientation of particles in the specimen, non-tilted and tilted datasets were acquired (Supplementary Table 1). Compared to the mature mtLSU[4,6], a large portion of density corresponding to interfacial rRNA is not observed in the consensus map owing to conformational heterogeneity of these elements (Fig. 5a). This region comprises helices H67-71 (domain IV) and H89-93 (domain V) of the 16S mt-rRNA, encompassing the peptidyl transferase centre (PTC). Additionally, this mtLSU intermediate contains density proximal to uL14m and bL19m corresponding to the MALSU1:L0R8F8:mtACP anti-association module (Fig. 5a and Supplementary Fig. 5). The presence of this module is indicative of particles that are not engaged in translation, as either biogenesis[21] or recycling[22] intermediates. Since the levels of the recycling factor MTRES1 (C6orf203) in fractions enriched in mtLSU particles from MRM2-depleted mitochondria were comparable to those observed in control cells (Supplementary Fig. 6), we interpreted the accumulated mtLSU species as biogenesis intermediates rather than products of defective, damaged or recycling ribosomes.

Focused classification of the intersubunit interface of these particles showed that this region is highly heterogeneous, with at least five distinct configurations present in the ensemble (Fig. 5b and Supplementary Fig. 7). The most incomplete

interface is present in state 1, where H67-71 (domain IV), the apex of H89 (domain V), and H90-93 (domain IV) are not observed. The base of H67 appears in state 2 as a highly disordered region at the base of the intersubunit interface. Furthermore, these two states present density protruding from the PTC, near the base of H89. In state 3, H92 (which contains the target of MRM2 – U3039) becomes slightly stabilised in a near-mature conformation by interaction with H90. In state 4, H89 and H93 become structured and H92 is further stabilised. With the organisation of H89-93, the configuration progresses to state 5, which presents H68-71 in their mature conformation and is thus the most complete and mature-like state; however, this is also the least populated state (~1% of total number of particles, Supplementary Fig. 7), possibly representing particles that were able to stochastically advance through the pathway without the aid of MRM2. Throughout these states, H89 changes from an outwardly rotated conformation with apical flexibility (states 1–3) to one where the whole helix is stabilised along the L7/L12 stalk (states 4 and 5) but its base is shifted to a different position relative to that in the mature state (Fig. 5c). Concomitantly to H89, the proximal H90-91 are also stabilised in their mature conformation (Supplementary Fig. 8a), and bL36m (absent in states 1–3) is present (Supplementary Fig. 8b). The MALSU1:L0R8F8:mtACP anti-association module is clearly present in all states and further classification schemes did not produce evidence in favour of the existence of particles lacking this component (Supplementary Fig. 9a).

To further evaluate structural heterogeneity in the dataset, we used a neural network-based approach—cryoDRGN[23]. Using information from whole particles, we corroborated that the intersubunit interface is the main source of structural variability in the studied ensemble of mtLSU particles (Supplementary Movie 1). In addition, this approach revealed particles with density connecting the base of H67 and the PTC, often with an extension to the space between the L1 stalk and the central protuberance (Supplementary Fig. 8c). In some cases, including states I and II (states derived from cryoDRGN outputs are indicated in roman numerals), this density could be traced into the back of the PTC, occupying a similar position to the protrusion observed in states 1 and 2. Nonetheless, some density corresponding to H92 is present in state II, providing some evidence that the protrusion does not contain this helix but may be composed of H90, H91, H93 or some combination of these, with an alternative conformation and possibly secondary structure. While H89 is flexible and outwardly rotated in states

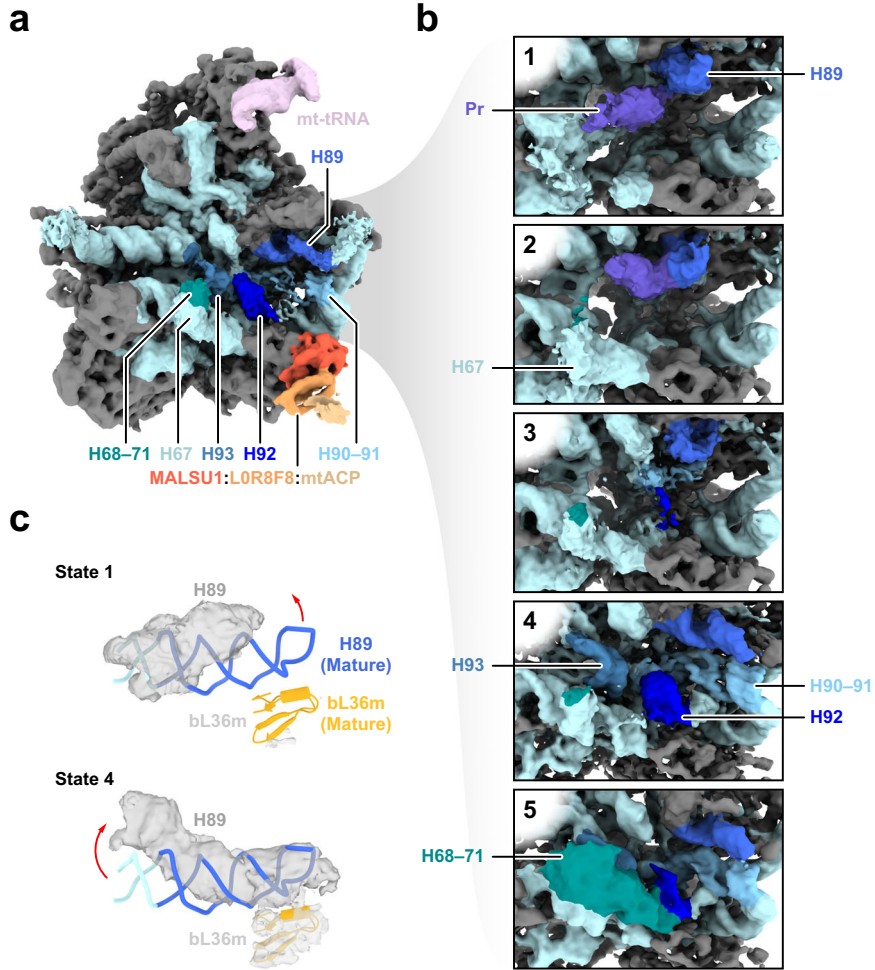

**Fig. 5 Structure of mtLSU assembly intermediates isolated in the absence of MRM2. a** Consensus map of the mtLSU intermediate viewed from the intersubunit interface. Relevant features are coloured. MRPs are represented in grey and 16S mt-rRNA in light blue. **b** Clarification of structural heterogeneity of the intersubunit interface by focused classification after masked signal subtraction. States/classes are numbered 1 to 5, corresponding to steps along a putative assembly pathway. Pr: protrusion. Models of relevant regions of states 1 and 4 are presented in Supplementary Fig. 8a. **c** Comparison between the conformation of H89 in state 1 (surface), state 4 (surface), and the mature mtLSU (ribbon). Rotation of the apex (state 1) and shift in the base (state 4) of H89 are represented by red arrows. Density corresponding to bL36m in each state is shown as surface. The conformation of H89 and density of bL36m in states 2 and 3 are similar to that modelled in state 1, and those in state 5 are represented by state 4. Density of bL36m is presented for states 1–5 in Supplementary Fig. 8b.

I and II, it appears in its mature conformation in state III, in which most of the interfacial RNA components (H80-H93) are in mature-like conformations. However, while H68-71 in state III are more ordered than in state 4, they do not reach the same extension as in state 5. Comparing these results with those obtained by focused classification, it was possible to place state I prior to state 3, state II between states 3 and 4, and state III between states 4 and 5.

Additional structural variation was observed in the central protuberance of mtLSU particles from cells devoid of MRM2 (Supplementary Fig. 9b). Most particles (~85%) lack mL40, mL46, mL48 and present a poorly defined structural mt-tRNA, similar to the consensus map. In fewer particles, the structural mt-tRNA appears stabilised and better resolved. Even though poorly defined, density for the missing proteins is observed in ~7% of the particles. However, this compositional heterogeneity does not reflect structural variations in other regions of the complex.

Taken together, these data show that MRM2 is involved in the late-stage assembly of mtLSU. However, whether the catalytic activity of MRM2 is required for this role remains to be established.

**The catalytic activity of MRM2 is dispensable for mtLSU biogenesis.** While acting as an assembly factor of the mtLSU, MRM2 is also a S-adenosyl methionine (SAM)-dependent 2′-O-ribose methyltransferase with its target on the 16S mt-rRNA. Therefore, we asked whether the catalytic activity of MRM2 is essential for mitoribosome biogenesis and/or if the MRM2 protein acts as a platform for local conformational/compositional changes upon binding to assembling mtLSU particles. The putative RNA binding site of MRM2 is composed by evolutionarily conserved surface residues such as K59, which are proximal to the SAM binding site and form an electropositive patch that stabilises the negative charge of the target. The SAM binding pocket is also lined by conserved residues, such as D154, which stabilises the methionine moiety while being positioned close to its Sδ atom[24] (Fig. 6a). Mutation of the catalytic residues K38 and D124 of *Escherichia coli* RlmE (equivalent to *Homo sapiens* MRM2 K59 and D154, respectively) has been shown to abolish its methyl-transferase activity[25]. Validation of the catalytic inactivity of the human variants was obtained by RNA LC-MS2, which identified 16S mt-rRNA oligonucleotides containing U3039 solely in its non-methylated state both in *MRM2* knock-out cells

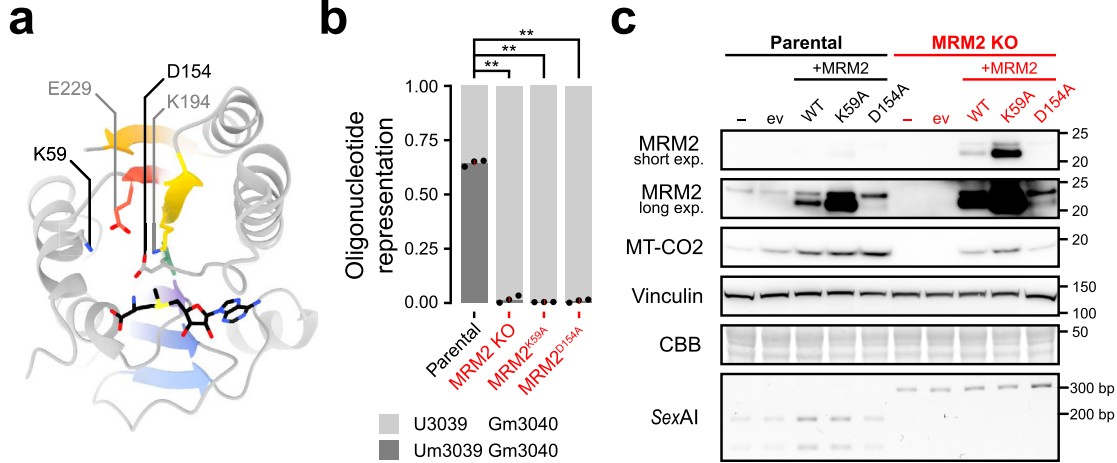

**Fig. 6 Catalytic mutants of MRM2 are able to restore mitochondrial translation. a** Structure of human MRM2 (PDB 2NYU)[69]. Evolutionarily conserved active site residues relevant for MRM2-mediated 2′-O-methylation are labelled and shown as sticks. S-adenosyl methionine (SAM) is represented as black sticks. The seven β-strand methyltransferase fold is coloured by strand, from N- (violet) to C-terminus (red). **b** Quantification of 2′-O-methylation of U3039 and G3040 by RNA LC-MS2. MRM2$^{K59A}$ and MRM2$^{D154A}$ correspond to samples from *MRM2* knock-out cells expressing those catalytic mutants. Statistical significance was assessed using two-tailed Student's *t* test (∗∗: $P \leq 0.01$; Parental vs *MRM2* KO $P = 0.0073$, Parental vs *MRM2*$^{K59A}$ $P = 0.0075$, Parental vs *MRM2*$^{D154A}$ $P = 0.0075$). **c**, Immunoblot evaluation of functional rescue of mitochondrial translation in cell lines complemented with wild-type (WT) MRM2, as well as catalytic mutant variants (K59A, D154A). ev: empty vector control. exp.: exposure. Molecular weights of protein standards are presented in kDa to the right of each blot. Coomassie brilliant blue (CBB) staining is shown as a loading indicator. For each cell line, electrophoretically separated *Sex*AI-digested amplicons of the genomic *MRM2* locus targeted for gene editing are presented. This experiment was replicated twice with similar results. Source data are provided as a Source Data file.

complemented with MRM2$^{K59A}$ and MRM2$^{D154A}$, as well as in the knock-out background (Fig. 6b and Supplementary Fig. 10). In control cells, ~70% of mtLSU-associated 16S mt-rRNA was present as Um3039/Gm3040, with the remainder ~30% representing the U3039/Gm3040 state; a state lacking the modification deposited by MRM3 was not detected.

Next, the steady-state level of MT-CO2 was used as a proxy (Fig. 3a, b) to assess the functional status of mitochondrial translation (Fig. 6c). As observed by the restoration of the expression of MT-CO2, both K59A and D154A MRM2 variants were able to rescue the defect in mitochondrial translation caused by knock-out of the endogenous gene despite their null methyltransferase activity. This led to the conclusion that the main contribution of MRM2 to mitochondrial translation is not its methyltransferase activity but its presence and role as an assembly factor, remodelling the intersubunit interface of the assembling mtLSU and thus contributing towards the mature conformation.

**Other mtLSU biogenesis factors are not redundant to MRM2**. Overexpression of certain ribosomal biogenesis factors has been reported to rescue the severe growth and 50S biogenesis defects of bacteria lacking the orthologue of MRM2 (RlmE/RrmJ/FtsJ/MrsF). This was the case of the GTPases ObgE/CgtA and EngA[26]. GTPases have been implicated in the biogenesis of the human mitochondrial ribosome, some of which orthologous to bacterial proteins. To test whether mitochondrial GTPases can rescue the MRM2 ablation phenotype in human, we complemented *MRM2* knock-out cells with GTPBP5/MTG2, GTPBP7/MTG1, and GTPBP10, which are known to be present in mitochondria and participate in mtLSU biogenesis[27–31]. However, despite the overexpression of these proteins, mitochondrial translation was not functionally restored, as indicated by the absence of MT-CO2 in the complemented cells (Supplementary Fig. 11), suggesting a nonredundant role of MRM2 in the assembly of mtLSU.

**MRM2 is required for organismal homeostasis**. Given the severe phenotype caused by ablation of MRM2 in cultured cells, as well as reports of patients harbouring variants in the coding gene presenting with a Mitochondrial Encephalomyopathy, Lactic Acidosis, and Stroke-like episodes (MELAS)-like pathology[13], we investigated the relevance of this protein for whole-organism and tissue-specific homeostasis in a *Drosophila melanogaster* model. Therefore, we knocked-down the *MRM2 D. melanogaster* orthologue *CG11447* (henceforth referred to as *DmMRM2*) by expressing an inducible (UAS-) RNAi construct driven by the ubiquitous promoter *da-GAL4* driver (Fig. 7a). Downregulation of DmMRM2 led to developmental delay, with few larvae progressing to pupae, and most of these dying at late pupal stage. In rare cases where eclosion was successful, adults were either trapped by the pupal case while emerging or immobilised on the food as they were too weak to escape. These adults showed anterior thoracic indentations alongside deformed wings and flattened abdomen (Fig. 7b). *DmMRM2* knock-down was lethal when performed under another ubiquitous driver (*act5C-GAL4*) or a pan-neuronal driver (*nSyb-GAL4*), further indicating the essentiality of this protein.

To evaluate the correlation between the lethal phase and requirements on mitochondrial function, the steady-state level of subunits of MRC complexes was assessed in different stages of development (Fig. 7c). Third instar larvae (L3) and young adults contrasted in the amount of these subunits, which is in line with their different metabolic needs[32,33]. This cellular and molecular shift occurred during pupation, where the steady-state levels of MRC subunits were progressively upregulated. Steady-state levels increased considerably during the second half of pupation, suggesting a stronger reliance of cells on mitochondrial respiration. This increase was less marked in *DmMRM2* knock-down animals, with levels of MRC subunits in late pupation stages being considerably lower when compared to matched control animals. Concurrently, the development of *DmMRM2* knock-down individuals is arrested at this stage, linking the observed

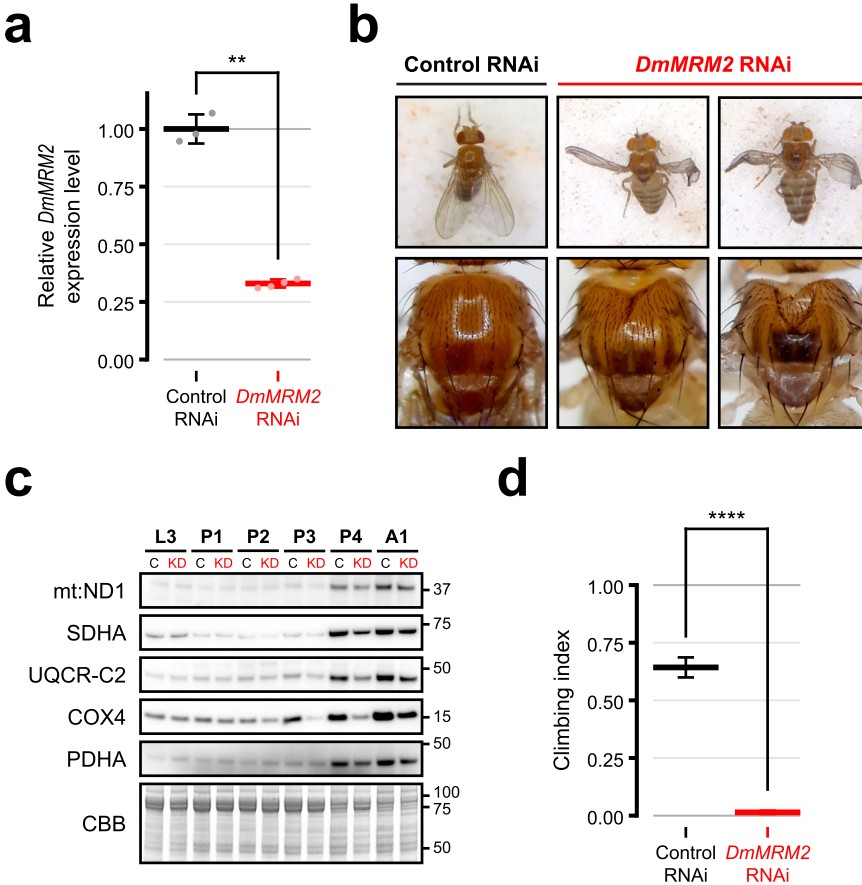

**Fig. 7 DmMRM2 is detrimental for the development of *D. melanogaster*, in particular towards the end stages of pupation. a** Levels of *DmMRM2* transcripts in L3 larvae ubiquitously expressing control ($n = 3$) or *DmMRM2* ($n = 4$) RNAi under the *da-GAL4* driver. Values are normalised by the level of *αTub84B* transcripts. Data are presented as individual datapoints and mean ± SD. Each datapoint was generated from three L3 larvae. Statistical significance was assessed using two-tailed Student's *t* test (∗∗: $P \leq 0.01$; $P = 0.0019$). **b** Dorsal view of whole-body and thorax of control and *DmMRM2* knock-down adult flies. **c** Immunoblot assessment of the steady state levels of MRC subunits over the course of pupation in control (C) and *DmMRM2* knock-down (KD) individuals. L3: third instar larvae; P1: day 1 pupae; P2: day 2 pupae; P3: day 3 pupae; P4: day 4 pupae; A1: adults after eclosion. Molecular weights of protein standards are presented in kDa to the right of each blot. Coomassie brilliant blue (CBB) staining is shown as a loading indicator. This experiment was replicated twice with similar results. **d** Startle-induced negative geotaxis (climbing) assay performed on adult male flies expressing control ($n = 70$) or *DmMRM2* ($n = 56$) RNAi under the pan-muscular *Mef2-GAL4* driver. Data are presented as mean ± SEM. Statistical significance was assessed using the Mann-Whitney test (∗∗∗∗: $P \leq 0.0001$; $P < 0.00001$). Control RNAi: *UAS-lacZ RNAi; da-GAL4* (transcript quantification, imaging, immunoblotting), *UAS-lacZ RNAi; Mef2-GAL4* (climbing assay). Source data are provided as a Source Data file.

lethality to the role of this protein in mitochondrial function and its relevance for cell and organismal homeostasis.

In order to investigate the impact of these molecular alterations at the whole organism level, and since muscular tissue is commonly affected in mitochondrial disorders[34], including the MELAS-like presentation of patients harbouring *MRM2* mutations[13], *DmMRM2* was knocked-down using a pan-muscular driver (*Mef2*-GAL4). While this did not lead to pupal lethality, adult flies presented considerably impaired locomotor ability in the startle-induced negative geotaxis assay, which monitors neuromuscular function (Fig. 7d). Taken together these data show that MRM2, through its involvement in mtLSU biogenesis, is indispensable for organismal homeostasis.

## Discussion

To understand the role of proteins associated with human pathogenesis, it is necessary to comprehend their function and the molecular mechanisms in which they are involved. In this study, we performed a multidisciplinary analysis of the mitochondrial role of MRM2, a protein that underlies the molecular basis of the MELAS-like syndrome in patients harbouring variants of the coding gene.

We demonstrate that all known 2′-*O*-methylations in human mitochondrial transcripts, G2815, U3039 and G3040, are introduced by MRM1, MRM2 and MRM3, respectively, and using transcriptome-wide approaches we detect no additional targets for these enzymes in mtRNAs (Fig. 1). As mentioned, MRM2 has been associated with human disease with a typical mitochondrial cytopathy presentation[13]; to study this in greater detail, we first generated a cellular knock-out model. Ablation of *MRM2* leads to mitochondrial dysfunction with severe reduction in cellular respiration and activity of MRC complexes containing mitochondrially-encoded subunits (Fig. 2). We show that this is caused by the impairment of mitochondrial translation, leading to reduced production of mtDNA-encoded proteins (Fig. 3, and Supplementary Figs. 2, 3). Consequently, this impacts on the structural integrity of the OxPhos system and thus explains its functional collapse. The defect in mitochondrial translation is associated with the large subunit of the mitochondrial ribosome, which RNA core element (16S mt-rRNA) is 2′-*O*-methylated by MRM2 in the A-loop residue U3039. Despite shown to be

dysfunctional in the absence of MRM2, mtLSU particles contain virtually all of their MRPs and accumulate in a near mature state (Fig. 4 and Supplementary Fig. 4).

The late stages of mtLSU assembly involve almost exclusively the remodelling of interfacial RNA elements of otherwise complete particles, as shown by recent structural studies[35–39]. Key players include the helicase DDX28, which dislocates the central protuberance to increase solvent exposure of the intersubunit interface, the MTERF4:NSUN4 complex, which holds H68-71 of domain IV of 16S mt-rRNA in an immature conformation to allow access of assembly factors to the PTC, and the GTPases GTPBP5, GTPBP6, GTPBP7 and GTPBP10, which coordinate the maturation of the PTC through a series of conformational rearrangements. In addition to these, the MALSU1:L0R8F8:mtACP module is bound in the reported assembly intermediates, avoiding their premature association with mtSSU particles and preventing the immature mtLSU from engaging in translation[21].

In the absence of MRM2, there is an accumulation of mtLSU particles containing the MALSU1 anti-association module and unstructured interfacial RNA components, similar to homeostatic assembly intermediates reported previously[21] (Figs. 4, 5). However, in contrast to homeostatic conditions, where these intermediates correspond to a relatively small portion of the observed mtLSU particles, this representation is virtually absolute in cells devoid of MRM2. Based on the features of the obtained intermediates, we hypothesise the following sequence of assembly steps, starting from a state lacking bL36m, with unstructured H67-71 (domain IV) and H90-93 (domain V, possibly in an alternative RNA structure): (1) stabilisation of the base of the intersubunit interface via H67; (2) initial folding of H92 by interaction with the base of H90; (3) folding of H90-93, inward rotation of H89 by interaction with H91 and concomitant structuring of the bL36m binding pocket, favouring its incorporation; (4) refolding of H68-71 via interaction with stabilised interface components, including H92. In the absence of MRM2, most particles accumulate before step (3) can occur, indicating that this protein contributes to the late stages of mtLSU biogenesis not only by stabilising its binding substrate, H92, in the proper folding configuration, but also by relocating H89 and H91. Although maturation of domain IV does not seem strictly dependent on the presence of MRM2, as evidenced by a low number of particles where these components are structured (Fig. 5b state 5 and Supplementary Fig. 7), it may be catalysed by the folding of proximal RNA helices and the presence of Um3039, which participates in the bacterial Um2552 (H92) – C2556 (H92) – U1955 (H71) interaction triad[40], corresponding to Um3039 (H92) – C1373 (H92) – U948 (H71) in human mtLSU, and may contribute to the positioning of G3040 by steric hindrance created by the 2′-methoxyl of Um3039.

Analysis of residual heterogeneity in the dataset (Supplementary Fig. 8c and Supplementary Movie 1) brought to light additional states of mtLSU particles, some of which recapitulate results obtained by conventional focused classification with signal subtraction (FCwSS), while others probe additional configurations in the mtLSU conformational space and/or assembly pathway. This enabled a clearer visualisation of density of unknown identity, with RNA helix-like features, protruding from the PTC and in the proximity of the unstructured H68-71. Furthermore, it added to the evidence against the existence of an appreciable number of mtLSU particles without the MALSU1:L0R8F8:mtACP module in the absence of MRM2.

It was shown that the bacterial ribosomal protein bL36 integrates into LSU particles late during assembly, which is triggered by the presence of the MRM2 orthologue, RlmE/RrmJ/FtsJ/MrsF[40]. Inspection of the mtLSU assembly intermediates described in the present work shows that incorporation of bL36m

is MRM2-independent (Fig. 5 and Supplementary Fig. 8), and most likely relies more directly on the acquisition of a proper folding of interfacial rRNA elements, namely H89, which in turn depends on the folding of H91 via stabilisation of H92, aided by MRM2.

Moreover, the final stages of the assembly of mtLSU particles are independent of the 2′-O-methylation of U3039 by MRM2 (Fig. 6). However, they depend on the transit through conformational states aided by a set of proteins, including MRM2. In the absence of this protein, mtLSU particles are still able to stochastically reach a near mature conformation, even though at a nominal rate. Nevertheless, since the conformation of these mtLSU is not similar enough to that seen in mature particles (displaced H89 base), they are not licenced to engage in translation, as seen by the persistence of the MALSU1:L0R8F8:mtACP module. Thus, the strict quality control of mtLSU assembly is fulfilled. This adds to previous evidence that the binding of assembly factors with RNA modification activity to assembling cytosolic ribosomes is frequently associated with quality-control checkpoints during the biogenesis of these complexes. However, the relevance of those enzymes, often evolutionarily conserved, is not related to the modification they introduce in rRNAs, but to their presence in cells[41].

Analysis of the methylation of 16S mt-rRNA nucleotides in MRM1-3 knock-out models showed that the 2′-O-methylation of U3039 by MRM2 depends on the prior modification of U3040, substrate of MRM3 (Fig. 1). This implies that MRM3 exerts its activity over the assembling mtLSU prior to MRM2, adding molecular evidence to support structural findings[35–39].

In bacterial systems, ablation of the MRM2 orthologue can be compensated by the overexpression of certain GTPases (ObgE/CgtA and EngA). The resulting bacterial ribosomal large subunits still lack 2′-O-methylated U2552 (equivalent to U3039 in human mitochondria) but are nonetheless able to engage in translation[26]. In S. cerevisiae, only a partial suppression in the thermosensitive loss of mtDNA of a Δmrm2 mutant strain was observed when a eukaryotic orthologue of ObgE, Mtg2p (MTG2/GTPBP5 in human), was overexpressed[26,42]. Although expression of human orthologues of mitochondrial GTPases did not complement the knock-out of MRM2 (Supplementary Fig. 11), catalytic mutants of this methyltransferase were able to functionally restore mitochondrial translation without recovering the modification of U3039 (Fig. 6), a process for which the mitochondrial system seems to lack redundancy.

We further tested the role of MRM2 in the model organism D. melanogaster, showing that its orthologue DmMRM2 is detrimental for development. This is especially relevant upon the shift from a larval glycolytic phenotype that supports rapid growth, to a more efficient respiratory phenotype in adults, which occurs during pupation[32,33]. It is possible that maternal deposition of factors and mitochondria contribute to the maintenance of mitochondrial function during larval stages when DmMRM2 is knocked-down. However, the turnover of OxPhos complexes, unmet by the lower rate of synthesis of their core subunits, as well as the increased reliance on OxPhos itself eventually tip the homeostatic balance during pupation towards an unbearable cellular burden (Fig. 7). Furthermore, DmMRM2 knock-down in D. melanogaster has the potential to be a model of mitochondrial dysfunction that does not target the OxPhos system directly but broadly affects its mtDNA-encoded components. This constitutes a useful tool for further investigation of fundamental questions, such as those related to differential effects of mitochondrial dysfunction associated with tissue specificity.

Our results identify a crucial point of control during the assembly of the mammalian mtLSU with mechanistic insights that enlighten not only the role of MRM2 in mitochondrial

translation, but also the phenotype caused by its disruption, and consequent implications at the multi-tissue level. Given the contraction in the number of modified residues in the mammalian mitochondrial ribosome and the knowledge on the enzymatic repertoire responsible for these modifications it will be important to determine the role of Um3039 in mitochondrial translation rate and fidelity as well as the essentiality of the remaining modifications and/or the involved enzymes in order to better understand mitoribosome biogenesis and mitochondrial translation.

## Methods

**Materials and resources**. Reagents (antibodies, chemicals, oligonucleotides, plasmids), deposited data, software/algorithms and equipment used/generated in this work are listed in Supplementary Table 2.

**Experimental models**. Unless stated otherwise, HEK 293T Flp-In T-REx and HEK 293T were maintained in high glucose DMEM with GlutaMAX and sodium pyruvate supplemented with 10% FBS, 1x penicillin-streptomycin. All cells were kept at 37 °C and in a 5% $CO_2$ atmosphere. Stocks used in this study are reported in Supplementary Table 2.

*Drosophila melanogaster* flies were maintained in a humidified temperature-controlled incubator set to 25 °C with 12 h/12 h light/dark cycles. Crosses and respective progenies were raised in a humidified incubator at 29 °C. Food consisted of agar, cornmeal, molasses, propionic acid and yeast. Stocks used in this study are reported in Supplementary Table 2.

**Knock-out cell line generation**. HEK 293T Flp-In T-REx cells were transfected with plasmids encoding the (+) and (−) ZFNs target to *MRM2* using the Cell Line Nucleofector Kit V with the A-23 program in the Nucleofector 2b device, according to the supplier's recommendations. After recovery, clonal lines were screened by immunodetection of MRM2 and subsequent DNA sequencing (amplification with MRM2 Fw and MRM2 Rv). The generated knock-out cells were cultured in medium supplemented with 20% FBS and 50 mg mL$^{-1}$ uridine.

**RNA extraction for library preparation**. RNA was extracted from parental and HEK 293T Flp-In T-REx cells in which *MRM1*, *MRM2* or *MRM3* was knocked-out. For each condition 3 biological replicates were obtained (one 15 cm plate each). Mitochondrial isolation and RNA extraction were conducted as previously described[43]. In brief, collected cells were resuspended in mannitol-containing mitochondria extraction buffer and incubated on ice for 20 min. Cells were then homogenised by running the suspension 15 times through a 25 G needle, followed by a series of low- and high-speed centrifugations. The resulting pellet, representing the mitochondria-enriched fraction, was used as input for RNA purification using TRI Reagent according to the manufacturer's instructions.

**Library preparation and sequencing**. Strand-specific libraries were generated on the basis of previously described protocols[15,16,43–45]. In brief, for each sample, 200 ng of mitochondria-enriched RNA was fragmented by mixing the RNA in bicarbonate buffer (pH 9.2) and heating for 5 min at 92 °C. Next, samples were subjected to FastAP Thermosensitive Alkaline Phosphatase and Turbo DNase treatments, followed by a 3′ ligation of RNA adaptor 1 using T4 ligase. Ligated RNA was separated into two identical pools and reverse transcribed using SuperScript III Reverse Transcriptase in two distinct concentrations of deoxyribonucleotide triphosphates (dNTPs; final concentration 2 μM and 500 μM), using the RT primer. As the used RT-enzyme stalls at 2′-O-methlated sites in low dNTP conditions, the first pool, subjected to 2 μM dNTPs was later used to obtain RT-stop signals[16]. The second pool, transcribed using 500 μM dNTPs, was used to obtain the background signal for RT-stops (elaborated in "Identification of putative 2′-O-methylation sites"), as well as to calculate cleavage protection scores ('score A')[15]. The resulting cDNA was subjected to a 3′ ligation with adaptor 2 using T4 ligase. The single-stranded cDNA product was then PCR amplified for 9-12 cycles. Libraries were sequenced on a NextSeq 500 platform generating short paired-end reads, ranging from 25 to 55 bp from each end.

**Identification of putative 2′-O-methylation sites**. A reference genome for alignment of reads was generated on the basis of chrM from the GENCODE release 32 (GRCh38.p13). Reads were mapped using the STAR aligner[46] with –alignIntronMax set to 1. To identify methylated sites, coverage and number of reads starting and ending at each position were counted using bam2ReadEnds.R[47,48].

Determination of RT-stop scores. Under limiting dNTP concentrations, the RT stops one nucleotide downstream of the methylated site. Thus, for each position, an RT-stop ratio was calculated by dividing the number of reads stopping at the downstream position by the coverage at that position. RT-stop ratio of low dNTP samples was then divided by the RT-stop ratio of the high dNTP sample, which resulted in a fold change RT-stop score representing enrichment of the 2′-O-methylated site in low dNTP versus

the background signal. The significance of changes in methylation level in wild-type and knock-down cells was assessed using Student's *t* test comparing the calculated fold change RT-stop scores in the three biological replicates of each condition.

Determination of cleavage protection scores. Cleavage protection scores were determined as in the calculation of 'score A' in[14]. The length of the flanking region used for calculation was set to six nucleotides, centred on the measured site. Scores were calculated only for positions with a median of >15 reads per position in the flanking 12 nucleotides surrounding the measured site. Significance of changes in methylation level in parental and knock-out cells was assessed using Student's *t* test comparing the calculated 'score A' in three biological replicates of each condition. The code for calculating RT-stop scores and cleavage protection scores is found in this paper's supplemental information (Data S1).

**Monitoring of cell proliferation**. Cells were plated in glucose-free DMEM supplemented with 10% FBS, 1x GlutaMAX, 1 mM sodium pyruvate, 1x penicillin-streptomycin and either 4.5 g L$^{-1}$ glucose or 0.9 g L$^{-1}$ galactose. Cell confluence was measured every 4 h using the Incucyte ZOOM High Definition Imaging Mode and analysed with the associated software.

**Assessment of mitochondrial respiration**. Cells were seeded and allowed to adhere for at least 6 h to Seahorse XF96 cell culture microplates previously coated with poly-L-lysine. Culture medium was removed and the cell layer washed with assay medium (DMEM without sodium bicarbonate, 4.5 g L$^{-1}$ glucose, 1x Gluta-MAX, 1 mM sodium pyruvate, 150 mM NaCl, pH 7.4). After equilibration in assay medium (1 h, 37 °C) with atmospheric $CO_2$, oxygen consumption rate (OCR) and extracellular acidification rate (ECAR) were measured in a Seahorse XF96 Extracellular Flux Analyzer, using sequential injections of 2 μM oligomycin, 0.35 μM BAM15, and 1 μM rotenone and 1 μM antimycin A. OCR and ECAR values were normalised by the total amount of DNA in each well, determined using the CyQUANT Cell Proliferation Assay kit.

**Spectrophotometric assessment of MRC activity**. The spectrophotometric measurement of the activity of MRC complexes was performed with small modifications to the protocol described in[49]. Measured activities are normalised to the activity of citrate synthase, and values are presented relative to those of the parental cell line.

**PAGE and immunodetection of proteins**. Proteins were denatured, separated by SDS-PAGE in Bolt 4-12% Bis-Tris Plus gels, and transferred onto nitrocellulose membranes via dry transfer. These were blocked with 5% non-fat milk in PBST for 1 h at room temperature, and then incubated over-night with primary antibodies at 4 °C. After three washes, membranes were incubated with the appropriate HRP-conjugated secondary antibody for 1 h at room temperature, washed and developed with ECL reagent using an Imager 680.

BN-PAGE was performed on whole cells following the protocol by Fernandez-Vizarra and Zeviani[50].

**In-gel assessment of MRC activity**. Following the resolution of whole-cell lysates by BN-PAGE, gels were incubated with the following solutions at room temperature: Complex I IGA solution (5 mM Tris-HCl pH 7.4, 1 mg mL$^{-1}$ nitro blue tetrazolium, 0.1 mg mL$^{-1}$ NADH), Complex II IGA solution (5 mM Tris-HCl pH 7.4, 1 mg mL$^{-1}$ nitro blue tetrazolium, 0.2 mM phenazine methosulfate, 20 mM succinate), Complex IV IGA solution (50 mM potassium phosphate buffer pH 7.4, 1 mg mL$^{-1}$ 3,3′-diaminobenzidine tetrahydrochloride hydrate, 1 mg mL$^{-1}$ cytochrome c, 24 U mL$^{-1}$ catalase, 75 mg mL$^{-1}$ sucrose). Upon appearance of coloured bands, gels were washed with water and scanned under white light.

**Labelling of *de novo* synthesised proteins**. Cells of interest were incubated twice in methionine/cysteine-free DMEM for 10 min, and then in methionine/cysteine-free DMEM supplemented with 5% dialysed FBS, 1x GlutaMAX, 1 mM sodium pyruvate, 96 μg mL$^{-1}$ L-cysteine for 20 min, after which 100 μg mL$^{-1}$ emetine were added. After 30 min, 100 μCi mL$^{-1}$ of [$^{35}$S]-L-methionine were added and labelling proceeded at 37 °C for 1 h. Cells were collected and washed with PBS before being lysed by addition of 0.1% DDM, 1.7 U μL$^{-1}$ benzonase and 1x cOmplete protease inhibitor cocktail. Sample components were separated in 10-20% Tris-glycine SDS-PAGE gels, which were stained with Coomassie Brilliant Blue before being dried. The autoradiogram was obtained by exposing the dried gel to a phosphor screen and developed using a Typhoon Biomolecular Imager. Densitometry profiles and baseline-corrected quantification were performed using ImageJ[51].

**Isolation of mitochondria**. Cells were detached from culture vessels and incubated on ice in hypotonic buffer (20 mM HEPES pH 7.8, 5 mM KCl, 1.5 mM MgCl$_2$, 2 mM DTT, 1 mg mL$^{-1}$ BSA, 1 mM PMSF, 1x cOmplete protease inhibitor cocktail) for 10 min. A Balch-type homogeniser was used to mechanically lyse cells by 5 passes through the chamber loaded with a 12 μm clearance ball bearing. After homogenisation, 2.5x MSH buffer (50 mM HEPES pH 8.0, 525 mM mannitol, 175 mM sucrose, 5 mM EDTA, 5 mM DTT, 2.5 mg mL$^{-1}$ BSA, 2.5 mM PMSF, 2.5x cOmplete protease inhibitor cocktail) was added. Debris was pelleted by

centrifugation at 1000 $g$ for 10 min, and crude mitochondria subsequently pelleted from the supernatant at 10,000 $g$ for 20 min. These were layered on top of a discontinuous density gradient (1.5 M, 1.0 M, and 0.5 M sucrose in 10 mM HEPES pH 7.8, 5 mM EDTA) and ultracentrifuged at 120,000 $g$ for 1 h. The mitochondrial layer was washed in 1x MSH and pelleted at 10,000 $g$. All manipulations and centrifugations were performed at 4 °C.

**Assessment of mitoribosome composition**. The procedure followed that described for quantitative analysis of density gradients by mass spectrometry (qDGMS) in[20]. In brief, cells were cultured in DMEM for SILAC, supplemented with 20% dialysed FBS, 50 mg mL$^{-1}$ uridine, 1x penicillin-streptomycin, 200 mg L$^{-1}$ L-proline, and either 0.398 mM L-arginine and 0.798 mM L-lysine, or 0.398 mM $^{13}C_6$,$^{15}N_4$-L-arginine and 0.798 mM $^{13}C_6$,$^{15}N_2$-L-lysine. Cell populations were allowed to expand separately in these media for at least 7 doubling periods, with frequent media exchange. Upon harvesting, cell populations were mixed 1:1 based on total protein content. Isolated mitochondria were lysed (50 mM Tris-HCl pH 7.4, 150 mM NaCl, 1 mM EDTA, 1% Triton X-100, 1x cOmplete EDTA-free protease inhibitor cocktail, 2 U μL$^{-1}$ SUPERase•In RNase inhibitor), clarified and loaded on a continuous 10-30% (w/v) sucrose density gradient (50 mM Tris-HCl pH 7.4, 100 mM NaCl, 20 mM MgCl$_2$, 1x cOmplete EDTA-free protease inhibitor cocktail) formed with a Gradient Station. Gradients were spun at 100,000 $g$ in a TLS-55 swinging bucket rotor for 135 min at 4 °C. Fractions were manually collected from the top of each gradient. Results for each mix were corrected for imperfect mixing by taking into consideration the median ratio of labels from mass spectrometric analysis of whole cell mix lysates.

**Protein mass spectrometry**. Liquid samples were precipitated with 20 vol. of cold ethanol and incubated at −20 °C for 16 h. Precipitates were collected by centrifugation, redissolved in 50 mM ammonium bicarbonate buffer (pH 8.0) and digested with trypsin (12.5 ng μL$^{-1}$, 37 °C, overnight). Peptide mixtures were resolved by reverse-phase UPLC on an EASY-nanoLC1000 and an Acclaim Pep-Map C18 column (50 μm x 150 mm, 2 μm, 100 Å, 300 nL min$^{-1}$) using an 84 min gradient of 5% to 40% acetonitrile with 0.1% formic acid, followed by an increase in acetonitrile concentration to 90% and re-equilibration with 5% acetonitrile, within 105 min. Peptides were analysed by positive ion electrospray mass spectrometry using a Q Exactive plus mass spectrometer and a routine to fragment and analyse the 10 most abundant multiply charged peptide ions each second. Full scan MS data (400 to 1600 $m/z$) were recorded at a resolution of 70,000 with an automatic gain control (AGC) target of $10^6$ ions and a maximum ion transfer of 20 ms. Ions selected for MS2 were analysed using the following parameters: resolution 17,500; AGC target of 5 ×104; maximum ion transfer of 100 ms; 2 $m/z$ isolation window; for HCD, a normalised collision energy 28% was used; and dynamic exclusion of 20 s. A lock mass ion (polysiloxane, $m/z$ = 445.1200) was used for internal MS calibration. Proteins were identified using the MaxQuant software package. Peptide information was filtered by Perseus, removing proteins only identified by site that matched a decoy database of random peptides and contaminants. Preliminary assessment of incorporation of heavy amino acids was performed by processing the peptide information of the heavy-only labelled samples using MaxQuant.

**Mitochondrial ribosome footprinting**. Cells were incubated with 200 μg mL$^{-1}$ chloramphenicol for 10 min and 200 μg mL$^{-1}$ cycloheximide for an additional 5 min. After washing with PBS, the monolayer was flash-frozen on liquid nitrogen and thawed in lysis buffer (20 mM Tris-HCl pH 7.5, 150 mM NaCl, 5 mM MgCl$_2$, 1 mM DTT, 1% Triton X-100, 200 μg mL$^{-1}$ chloramphenicol, 200 μg mL$^{-1}$ cycloheximide). Half of the clarified lysates was incubated with 375 mU μL$^{-1}$ RNAse I at 28 °C for 30 min, and monosomes recovered by ultracentrifugation in a continuous 10-30% (w/v) sucrose density gradient containing chloramphenicol and cycloheximide. After 40 mU μL$^{-1}$ proteinase K incubation at 42 °C for 30 min, ribosome-protected fragments were extracted in acid phenol-chloroform. Total RNA was extracted from the other half using TRIzol LS, DNA degraded using TURBO DNase and mt-rRNA depleted using the Ribo-Zero Gold kit before alkaline fragmentation (45 mM sodium bicarbonate, 5 mM sodium carbonate, 1 mM EDTA, 95 °C, 15 min). Both RNA sample sets were subjected to size selection (25 nt to 35 nt) after TBE-Urea PAGE. Following, RNA was extracted from gel pieces (300 mM sodium acetate pH 5.5, 0.25% SDS, 4 °C, 18 h) and standard library preparation was carried out using the TruSeq Small RNA Library Preparation kit (Illumina). Libraries were sequenced (50 bp single read) in the CRUK Cambridge Institute Genomics Core using a HiSeq 4000 system. Adaptor sequences were trimmed from reads using FASTX-Toolkit (http://hannonlab.cshl.edu/fastx_toolkit/), trimmed reads mapping to nuclear-encoded ribosomal RNA (rRNA) were discarded, and the remaining reads were mapped sequentially to mitochondrial rRNA (mt-rRNA); mitochondrial transfer RNA (mt-tRNA); and mitochondrial messenger RNA (mt-mRNA) using bowtie version 1[52], with parameters -v 2 --best. To normalise for library size, read counts per mitochondrial gene were expressed as reads per million (RPM) mapping in the positive-sense orientation to nuclear-encoded mRNA (from NCBI RefSeq) in each library. When calculating mitochondrial gene ribosomal occupancy and translation efficiency (i.e. RiboSeq RPM/RNASeq RPM), overlapping CDS regions in ATP8/ATP6 and ND4L/ND4 were excluded due to ambiguous assignment. Additionally, reads with 5′ ends mapping within 15 nt of the start codon or 45 nt of the stop codon of each gene were excluded to avoid counting ribosomes paused during initiation or termination of translation.

**Purification of mtLSU**. Mitochondria were purified from *MRM2* knock-out cells as described in "Isolation of mitochondria", and lysed in 20 mM HEPES, 300 mM KCl, 50 mM MgCl$_2$, 1 mM DTT and 1% Triton X-100 supplemented with RNase and EDTA-free protease inhibitors. The clarified lysate was loaded on a continuous 10–30% (w/v) sucrose density gradient (20 mM HEPES pH 7.6, 300 mM KCl, 5 mM MgCl$_2$, 1 mM DTT) and ultracentrifuged at 100,000 $g$ for 135 min. Fractions enriched in mtLSU were pooled and ultracentrifuged at 625,700 $g$ for 35 min. The obtained pellet was resuspended in lysis buffer without Triton X-100 and loaded on a second continuous 10-30% (w/v) sucrose density gradient, which was ultra-centrifuged at 100,000 $g$ for 12 h. Fractions were automatically collected using an ÄKTAprime plus system with a 60% (w/v) sucrose chase solution. The absorbance profile at 260 nm was monitored and fractions enriched in mtLSU were pooled. Ribosomal particles were pelleted at 418,000 $g$ for 2 h. The pellet was washed in buffer without sucrose or DTT, resuspended, briefly centrifuged to remove aggregates, and used for cryoEM grid preparation. All manipulations and centrifugations were performed at 4 °C.

**CryoEM grid preparation and data collection**. Quantifoil holey carbon R2/2 mesh 300 copper grids were glow discharged at 20 mA for 60 s. Sample grids were prepared using a Vitrobot Mark IV set to 95% relative humidity and 4 °C. Purified mtLSU (~0.2 μg μL$^{-1}$ total protein) was applied to the carbon-coated side of grids (3 μL), blotted (15 s waiting time, 2.5 s blotting time, 0 blotting force) and plunge frozen in liquid ethane.

Data were collected at the cryoEM facility of the Biochemistry Department (University of Cambridge) on a 300 keV Titan Krios equipped with a Falcon 3EC direct electron detector in integrating mode. Due to preferential orientation of the particles, a second dataset was collected where the grid was tilted by 20°, as suggested by cryoEF[53] using data from the processed non-tilted dataset. A total of 10,078 and 10,401 micrographs were acquired for the non-tilted and tilted datasets, respectively. In both cases a total dose of ~52 e$^-$ Å$^{-2}$ was distributed over 18 frames, with an exposure time of 600 ms; C2 and objective apertures were 70 μm and 100 μm, respectively (Supplementary Table 1).

**CryoEM data processing and model building**. Acquired data were processed (Supplementary Fig. 5) using RELION-3.1[54,55]. Beam-induced motion correction was performed using MotionCor2[56] and CTF estimation using CTFFIND-4.1[57]. Particles were picked using the Laplacian of Gaussian (LoG) filter and imported into cryoSPARC[58] where an initial reference was generated ab initio and used for clearing the stack by 3D heterogeneous refinement. Clean particle stacks were processed (3D auto-refinement, CTF refinement, Bayesian polishing) individually with RELION-3.1 and then used for 3D classification. Particles from both datasets that led to the reconstruction of maps with detailed features were merged to generate a partial consensus map (846,646 particles); all particles were used to generate a full consensus (1,191,870 particles). Focused 3D classifications (Supplementary Figs. 7 and 9, and Supplementary Tables 3 and 4) were performed using RELION-3.1 by subtracting signal outside a mask over the regions of interest (FCwSS) and classifying particles without updating alignments; classes of interest were further 3D auto-refined. Masks used for FCwSS are presented in Supplementary Fig. 7, 9.

Full models of states 1 and 4 (Fig. 5) were built and refined. The structure of a previously reported mitochondrial assembly intermediate (PDB: 5OOL) was used as a reference model. This model was first fit into the density using Chimera[59]. Iterative cycles of model building and refinement of ribosomal proteins and rRNA were performed using *Coot*[60] and PHENIX real-space refinement[61], respectively. The model was validated with MolProbity[62] and the PHENIX suite[61]. Model building statistics are presented in Supplementary Table 3.

Heterogeneity analysis was performed using cryoDRGN[23]. A 1,191,870 particles stack, downsampled to 256 px (1.49 Å px$^{-1}$) was used, alongside poses and CTF parameters extracted from the reconstructed full consensus (Fig. 5a and Supplementary Fig. 5), to train the variational auto-encoder for 25 epochs with a 10D latent variable and a 1024×3 architecture. The distribution of latent encodings was visualised with the built-in 2D uniform manifold approximation and projection (UMAP) and $k$-means clustering was performed with $k$ = 30. A trajectory connecting the cluster centres in latent space was determined and volumes of anchor points along that path generated "on-data" by evaluating the trained decoder with their latent variable values. Additional cluster centres were determined by selecting a region of interest in the UMAP representation using the cryoDRGN-generated Jupyter Notebook, determining the median coordinates for the selected datapoints and identifying the closest datapoint that represents a particle from the dataset.

Figures of maps and models were generated using ChimeraX[63].

**Lentivirus production and stable cell line generation**. Lentivirus were produced in HEK 293 T transiently co-transfected with pWPXLD:IRES:PuroR containing the transgene of interest, psPAX2 and pMD2.G using FuGENE 6 transfection reagent according to the manufacturer's instructions, using a DNA mass to transfection

reagent volume ratio of 0.33. Lentiviral supernatants collected after 48 h were spun, filtered and split equally among cells of interest. Cells that successfully integrated the transfer plasmid were selected with 1 µg mL$^{-1}$ puromycin.

To verify the genetic background of cell lines, the genomic region modified to knock-out *MRM2* was amplified and digested with *Sex*AI. Reaction products were separated by electrophoresis in 1% agarose TBE gels.

**RNA oligonucleotide mass spectrometry**. Large mitoribosomal subunits were purified from cultured cells using a continuous sucrose density gradient as described in "Purification of mtLSU". Subsequently, RNA was extracted from selected fractions using TRIzol LS and screened for purity and concentration using a 2200 TapeStation system. Oligonucleotides were prepared from RNA using RNAse T1 and chromatographically separated by ion pair reverse phase chromatography (200 mM HFIP, 8.5 mM TEA in water as eluent A, and 100 mM HFIP, 4.25 mM TEA in methanol as eluent B). The oligonucleotides were resolved by a non-linear gradient of 2.5% to 20% B at 200 nL min$^{-1}$ on Acclaim PepMap C18 solid phase and characterised by negative ion tandem LC-MS in a Q Exactive HF hybrid quadrupole-orbitrap. Data were collected in data-dependent acquisition mode, with full scan MS1 data acquired between 700 and 3500 *m/z*. The top five ions generating signals with the most intense signals were selected for fragmentation and subsequent MS2 characterisation. Tandem MS data were analysed using the OpenMS software suite[64], and custom R scripts. Oligonucleotide database searching was performed using NucleicAcidSearchEngine, according to parameters given in[65]. Briefly, isotopologues between the monoisotopic and +4 peak were considered for assignment with an accuracy of 3 ppm at MS1 and MS2, and single sodium and potassium ions were considered as adducts, in addition to single methylation of each nucleobase as a variable modification. Label-free quantification of oligonucleotide precursors was initially assessed using an updated version of FeatureFinderID[66] and confirmed by manual MS1 peak integration.

**Quantification of transcript levels**. Levels of *CG11447* (*DmMRM2*) and *αTub84B* transcripts were determined by RT-qPCR. Total RNA was extracted from 3 larvae using TRIzol reagent, genomic DNA removed with TURBO DNA-free Kit, and reverse transcribed with Maxima H Minus cDNA Synthesis Master Mix. The reaction products were amplified with PowerUp SYBR Green Master Mix using primer pairs targeting *DmMRM2* (DmMRM2 Fw and DmMRM2 Rv) and *αTub84B* (αTub84B Fw and αTub84B Rv). qPCR was performed in a QuantStudio 3 Real-Time PCR System. No-RT and no-template control reactions were included. The specificity of primer pairs was evaluated by the melting profile, and the amplification efficiency was 1.93 and 1.95 for the pair targeting *DmMRM2* and *αTub84B*, respectively. Transcript levels were determined using the comparative Ct method with amplification efficiency correction and normalised for the average of *αTub84B*[67].

**Locomotor assay**. The startle-induced negative geotaxis (climbing) assay was performed using a counter-current apparatus. Briefly, 2 days old adult male flies were placed into the first chamber, tapped to the bottom, and given 10 s to climb a 10 cm distance. This procedure was repeated five times (five chambers), and the number of flies that has remained into each chamber counted. The weighted performance was normalised for each genotype to the maximum possible score and expressed as climbing index[68].

**Quantification and statistical analysis**. Individual experimental values with average, mean ± SD or median values are presented. Statistical parameters, tests and significance, as well as number of replicates (*n*) can be found accompanying the respective data in figure legends.

**Reporting summary**. Further information on research design is available in the Nature Research Reporting Summary linked to this article.

## Data availability
Sequencing data generated in this study have been deposited in the GEO and ArrayExpress databases under accession codes GSE179085 and E-MTAB-11292, respectively. CryoEM maps generated in this study have been deposited in the EMDB database under accession codes EMD-13965, EMD-13962, EMD-13963, EMD-13967 and EMD-13966. Atomic models generated in this study have been deposited in the PDB database under accession codes 7QH6 and 7QH7. The source data underlying all figures and supplementary figures are provided as a Source Data file. Any additional information required to reanalyse the data reported in this paper is available from the lead contact upon request. Source data are provided with this paper.

## Code availability
Original scripts used in the analysis of 2′-*O*-methylation of mitochondrial transcripts are presented in Supplementary Data 1. Custom R scripts used for RNA mass spectrometry analysis are property of Storm Therapeutics.

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

## Acknowledgements

The authors would like to acknowledge Dima Chirgadze (cryoEM facility, University of Cambridge) for assistance with grid screening and acquisition of electron microscopy data, Michael Harbour, Shujing Ding and Ian M. Fearnley (Mass Spectrometry facility, MRC MBU, University of Cambridge) for their help in the proteomics analysis, Joanna Rorbach for the provision of the MRM3 knock-out cell line, Merlin Hartley and Andrew Raine for IT support, and the members of the MRC MBU Mitochondrial Genetics group and Daniel Grba for insightful discussions. This work was supported by core funding from Medical Research Council UK (MC_UU_00015/4, MC_UU_00015/6), Fundação para a Ciência e a Tecnologia (PD/BD/105750/2014) to P.R.-G., a specialist Programme from Blood Cancer UK (12048), the Kay Kendall Leukaemia Fund, the UK MRC (MR/T012412/1), a Wellcome Trust strategic award to the Cambridge Institute for Medical Research (100140), a core support grant from the Wellcome Trust and MRC to the Wellcome Trust—MRC Cambridge Stem Cell Institute, the Cambridge National Institute for Health Research Biomedical Research Centre (BRC-1215-20014) and the European Cooperation in Science and Technology (COST) Action CA18233, "European Network for Innovative Diagnosis and treatment of Chronic Neutropenias, EuNet INNOCHRON" to S.P., K.C.D., and A.J. Warren, and European Commission under "Marie Skłodowska-Curie Actions", Individual Fellowship—Reintegration Panel (Mitobiopath-705560) and Italian Minister of University and Research—Rita Levi Montalcini Program to C.G.

## Author contributions

P.R.-G. and M.M. planned and designed experiments. A.S.-C. and S.S. performed and analysed the transcriptome-wide methylation screening. C.G. performed spectrophotometric assays of MRC activity. A.D., A.E.F. and L.V.H. processed mitoribosome footprinting sequencing. J.F.R. and B.A. performed RNA mass spectrometry. P.R.-G. performed biochemical purifications, K.C.D. collected cryoEM data, P.R.-G. performed image processing and reconstructions with participation of S.P. and K.D, and S.P. built atomic models. L.M.-F. and P.R.-G. performed experiments in the *Drosophila* model. P.R.-G. performed the remaining experiments. P.R.-G. and M.M. drafted the manuscript. M.M., B.A., A.E.F., A.J. Whitworth, A.J. Warren and S.S. supervised the study. All authors revised the manuscript.

## Competing interests

J.F.R. and B.A. are employees of STORM Therapeutics Ltd. M.M. is a founder, shareholder and member of the Scientific Advisory Board of Pretzel Therapeutics, Inc. The remaining authors declare no competing interests.
