## [Peer Review File · Nature Communications]

A late-stage assembly checkpoint of the human mitochondrial ribosome large subunitREVIEWER COMMENTS

Reviewer #1 (Remarks to the Author):

The manuscript by Rebelo-Guiomar reports the results of an original investigation poised at addressing the late maturation steps of the mitochondrial ribosomal large subunit (mt-LSU). Through a transcriptome-wide analysis, the authors confirm that the 2'-O-methylation of the mt-LSU rRNA occurs through three methyltransferase enzymes, MRM 1, 2 and 3 (thanks to an elegant transcriptome-wide analysis based on a combination of two high throughput sequencing methods, RiboMeth-Seq and 2OMe-Seq), with the modification installed by MRM 2 and 3 being interdependent.

Importantly, the authors investigate the essentiality of MRM2, as several human pathologies were revealed to be related to mutations in the latter. They find out that its absence, independently of its enzymatic activity, could lead to mitochondria related developmental arrest in *D. melanogaster*.

It is interesting to mention that the authors were successful in demonstrating that the ablation of MRM2 doesn't lead to a codon-specific stalling or any other specific deficiency in a particular translation step, but to global significant reduction of mitochondrial translation. Indeed, the authors observed a reduced occupancy of mitoribosomes on the mt-mRNA via high throughput mitoRibo-Seq upon ablation of MRM2, while the distribution of mitoribosome on the mt-mRNA remained unchanged.

Using qDMGS, the authors established the accumulation of the mt-LSU or the near-mature mt-LSU upon the depletion of MRM2. Using cryo-EM, the authors were able to investigate the effect of the absence of the mt-LSU methylations and show that mRNA becomes flexible/disordered at several regions, in addition of presenting a partial occupancy for the mitoribosomal protein bL36m. The cryo-EM analysis shows the existence of at least 5 different maturation intermediates with different intersubunit interface configurations.

Surprisingly, the authors reveal that the methyltransferase activity of MRM2 is not essential to the biogenesis of the mt-LSU. Instead, the authors conclude that MRM2 is essential as an assembly factor that remodels the rRNA during a late stage of the mt-LSU maturation. This finding is literally amazing and merits to be highlighted!

Finally, the authors reveal that the over expression of other assembly factors doesn't restore normal mt-LSU maturation, so there is no functional redundancy of other maturation/assembly factors to MRM2.

The authors conclude on the presence of an essential quality control step during the mt-LSU maturation/assembly.

The manuscript is well written, and the figure are clear and justified. The overall quality of the reported study is remarkable and reports a wealth of experiments. The Methods section appears to be sufficient and experiments are well detailed.

I only have two minor comments:

1- The authors show through high throughput mitoRibo-seq that the impairment results from a global strong reduction of mitochondrial translation. mitoRibo-Seq shows a deficit in mitoribosome occupancy along the mt-mRNA. However, although I fully agree with the interpretation of the authors, this method doesn't measure directly mitochondrial translation but rather the positions/quantity of mitoribosomes on the mt-mRNA. Indeed, one would expect the activity of mitoribosomes to be altered in the absence of this methylation. For example, is it possible that the translation rate or fidelity of the mitoribosome declines in the absence of the U1369 methylation? These two aspects are independent of the occupancy deficit that the authors highlight with the mitoRibo-Seq experiment... Could the authors comment please?

2- The authors write that in state 3 only, H93 is relatively stabilized by interaction with H90. However, from Figure 5, state 3 show nearly no H92, perhaps the authors mean state 4?

Reviewer #2 (Remarks to the Author):

It was a pleasure to read the study by Rebelo-Guiomar et al., who investigated the molecular function and the physiological role of MRM2 in more detail. First, they generated cell lines deficient in the three 2'-O-methyltransferases MRM1, MRM2 and MRM3. Their genome-wide analysis suggests that 2'-O-methylations are only present in the 16S rRNA and that the modifications introduced by MRM2 and MRM3 are interdependent. In the absence of MRM2 mtLSU assembly is stalled at late maturation states accumulating five intermediates, which the authors resolved by cryo-EM. Most interestingly, although mtLSU maturation depends on MRM2, its methyltransferase activity is dispensable suggesting a second role as an assembly factor by remodeling the ribosomal interface side. The study is not only a good complementation to the previous reported mtLSU assembly structures, it also addresses the physiological role of MRM2 and its implications in human mitochondrial diseases.

The experiments performed in this study are well designed and show high quality data. Hence, this manuscript is suitable for publication in Nature Communications. However, I have a few minor issues, which the authors may want to consider:

1) The majority of structural studies follows the ribonucleotides numbering according to their relative position in the mitochondrial genome. The authors may consider adjusting this in their study. (G1145 = G2815; U1369 = U3039; G1370 = G3040)

2) Supplementary Fig. 1: The level of MRM1 decreases in MRM2 KO and MRM2 is reduced in MRM3 KO. Could the authors comment on this? It might be that the reduced 2'-O-methylation of U1369/U3039 in MRM3 KO is caused by the reduced levels of MRM2.

The authors included rescue experiments for MRM2 KO, but similar experiments for MRM1 and MRM3 are missing. This should be included to confirm the specificity of the KO.

3) Fig. 3b: TOM22 seems to be strongly increased, while the loading (beta-actin) seems to be fine. Is there an increase in mitochondrial mass in MRM2 KO and how can this be explained?

4) Fig. 4: It would also be great to include a western blot for the density gradient. From the graph in Fig. 4a it is not clear whether 55S monosomes are still formed in the KO. Would fraction 10 correspond to 55S? In addition, the authors could include some assembly factors in their gradient analyses e.g. GTPBP5, which seems to bind downstream of MRM2 to the mtLSU, or other GTPases. Although structurally not resolved, mtLSU intermediates accumulating in MRM2 KO might have other assembly factors bound. Thus, further biochemical analyses might be informative.

5) Page 9: This is a strange sentence: "We set out to investigate whether the relevance of MRM2 for mitoribosome biogenesis is due to the chemical modification it introduces in U1369 or to MRM2 acting as a platform for local conformational/compositional changes once bound to assembling mtLSU particles." Please, rephrase.

6) Supplementary Fig. 11: Although this blot is not of good quality, one could still conclude that the overexpression of mitochondrial GTPases affects mitochondrial translation as COX2 is decreased in the parental background if GTPBP5, -7, -8 or -10 are elevated. This is also in agreement with recent studies showing that overexpression of mitochondrial GTPases reduces mitochondrial translation as they act as anti-association factors preventing subunit joining (Maiti et al., 2020; Lavdovskaia et al., 2018). This might also be one of the reasons why their overexpression does not rescue the MRM2 KO phenotype. Maybe worth to include in the discussion.

Reviewer #3 (Remarks to the Author):

This manuscript addresses specific aspects of human mitochondrial ribosome biogenesis.

In particular, the authors are studying the involvement of MRM2, a 2'-O methyltransferase active on the mitochondrial large ribosomal subunit RNA, the 16S.

Briefly, the authors start by assigning a single 2'-O methylated residue of 16S to each of MRM1, MRM2, and MRM3. The remainder of the work focusses on MRM2, which is deleted in HEK293T (kidney) cells,

and depleted in flies. MRM2 was known to be associated to human MELAS-like syndrome which affects primarily the brain, the nervous system, and the muscles. In human kidney cells, the gene is not essential and delete cells are defective for mitochondrial function, mitochondrial translation, and (partially) for large ribosomal subunit assembly. In fly, the protein is indispensable for homeostasis, which really questions the pertinence of kidney cells as model (as I understand, this organ is not particularly affected in MELAS-like syndrome).

The manuscript has been prepared with great care and contains a lot of high-quality high-resolution data. Regretfully, the biological insight is too limited. Essentially, what the authors have shown is that cells deprived of a ribosome assembly factor don't make ribosome well, and consequently loose translation and function.

The authors are proposing the MRM2 is part of a quality control mechanism during late stage of mitochondrial ribosome biogenesis, but it is not entirely clear how this works? Indeed MRM2 is not essential in HEK293T cells, implying mitochondrial ribosomes are made in its absence. In fact *mrm2* $-/-$ cells are producing two types of ribosomes: 1) unmodified mature ribosome engaged in translation and sustaining cell growth, and 2) stalled large ribosomal subunits, which are translationally inactive because they have retained assembly factors masking functional sites (interface). The question is to know why sometimes the assembly process is halted, and why sometimes it goes to fruition. Another important question to answer is what is the respective amounts of mature ribosomes versus stalled precursors produced. Also important would be to know by how much production of mature ribosome is reduced (this is particularly important because ribosome profiling illustrates very well that translationally active ribosome translate indifferently from wild-type control ribosomes.)

With so many open questions, and the limited biological insights offered, I regret I cannot recommend publication of this manuscript in Nature Communications.

Other comments:

-to bring the MRM2 results in perspective, data on MRM1 and MRM3 would have been really very useful (in particular since all three modify the same ribosomal RNA).

-Unlike stated in the Discussion, the authors have not established that the catalytic function of MRM2 is not involved in ribosomal subunit assembly (they acknowledge this in the Results section; but somehow they don't in the Discussion). What they have shown, using rescue constructs, which have their own limitations, is that translation of select products resume, which is not the same (especially considering that a yet-to-be-determined level of mature ribosomes are produced in the delete cells...).

-Why is there so much heterogeneity at the interface on stalled precursor particles in the mutant (at least 5 conformations, if I understand well). Why is it so? Are these five, and possibly more conformations, similarly represented in the population of particles? Or are some more abundant than others?

-Why is it that the MRM2-mediated modification depends upon the MRM3-mediated modification but that the contrary is not true? At least, it's not because the metabolic stability of MRM2 is affected in *mrm3* $-/-$ cells (the authors show this but do not comment on it, they could). Is it because cells lacking MRM3 are also defective for ribosome biogenesis, may be at an earlier stage?

Reviewer #1

The manuscript by Rebelo-Guiomar reports the results of an original investigation poised at addressing the late maturation steps of the mitochondrial ribosomal large subunit (mt-LSU). Through a transcriptome-wide analysis, the authors confirm that the 2'-O-methylation of the mt-LSU rRNA occurs through three methyltransferase enzymes, MRM 1, 2 and 3 (thanks to an elegant transcriptome-wide analysis based on a combination of two high throughput sequencing methods, RiboMeth-Seq and 2OMe-Seq), with the modification installed by MRM 2 and 3 being interdependent.

*Importantly, the authors investigate the essentiality of MRM2, as several human pathologies were revealed to be related to mutations in the latter. They find out that its absence, independently of its enzymatic activity, could lead to mitochondria related developmental arret in *D. melanogaster*.*

It is interesting to mention that the authors were successful in demonstrating that the ablation of MRM2 doesn't lead to a codon-specific stalling or any other specific deficiency in a particular translation step, but to global significant reduction of mitochondrial translation. Indeed, the authors observed a reduced occupancy of mitoribosomes on the mt-mRNA via high throughput mitoRibo-Seq upon ablation of MRM2, while the distribution of mitoribosome on the mt-mRNA remained unchanged.

Using qDMGS, the authors established the accumulation of the mt-LSU or the near-mature mt-LSU upon the depletion of MRM2. Using cryo-EM, the authors were able to investigate the effect of the absence of the mt-LSU methylations and show that mRNA becomes flexible/disordered at several regions, in addition of presenting a partial occupancy for the mitoribosomal protein bL36m. The cryo-EM analysis shows the existence of at least 5 different maturation intermediates with different intersubunit interface configurations.

Surprisingly, the authors reveal that the methyltransferase activity of MRM2 is not essential to the biogenesis of the mt-LSU. Instead, the authors conclude that MRM2 is essential as an assembly factor that remodels the rRNA during a late stage of the mt-LSU maturation. This finding is literally amazing and merits to be highlighted!

Finally, the authors reveal that the over expression of other assembly factors doesn't restore normal mt-LSU maturation, so there is no functional redundancy of other maturation/assembly factors to MRM2.

The authors conclude on the presence of an essential quality control step during the mt-LSU maturation/assembly.

The manuscript is well written, and the figure are clear and justified. The overall quality of the reported study is remarkable and reports a wealth of experiments. The Methods section appears to be sufficient and experiments are well detailed.

We are grateful for the supportive comments, which state that the manuscript is of “*remarkable quality*”, with some of the key finding being referred to as “*amazing*”.

I only have two minor comments:

1- The authors show through high throughput mitoRibo-seq that the impairment results from a global strong reduction of mitochondrial translation. mitoRibo-Seq shows a deficit in mitoribosome occupancy along the mt-mRNA. However, although I fully agree with the interpretation of the authors, this method doesn't measure directly mitochondrial translation but rather the positions/quantity of mitoribosomes on the mt-mRNA. Indeed, one would expect the activity of mitoribosomes to be altered in the absence of this methylation. For example, is it possible that the translation rate or fidelity of the mitoribosome declines in the absence of the U1369 methylation? These two aspects are independent of the occupancy deficit that the authors highlight with the mitoRibo-Seq experiment... Could the authors comment please?

We applied mitoRibo-seq on earlier stages of the presented study, following the observation of a global reduction of mitochondrial translation rates as revealed by *de novo* metabolic labelling in MRM2-deficient cells. Then with mitoRibo-seq, we intended to analyse the stage at which mitochondrial translation is perturbed, assuming that it could be initiation, elongation or termination. Instead, mitoRibo-seq revealed a global reduction of the mitochondrial ribosome occupancy and this led to the hypothesis of incomplete assembly of the large subunit when MRM2 is absent. With the data in hand and available tools, we cannot precisely assess translation rate or fidelity of the mitoribosome in the absence of the U1369 methylation. To address the reviewer's point, we added a comment in the Discussion section (page 16): *"It will be important to determine the role of Um3039 in mitochondrial translation rate and fidelity as well as essentiality of the remaining modifications and/or the involved enzymes in order to better understand mitoribosome biogenesis and mitochondrial translation."*

2- The authors write that in state 3 only, H93 is relatively stabilized by interaction with H90. However, from Figure 5, state 3 show nearly no H92, perhaps the authors mean state 4?

We interpret that the reviewer means H92 and refers to the following sentence: *"It is only in state 3 that H92 (where U1369 is located) is slightly stabilised in a near-mature conformation by interaction with H90."*

Unlike states 1 and 2, states 4 and 5 present a clear density corresponding to H92. In state 3, while not completely resolved, there is density in the region occupied by H92. The same applies to H90, for which some density is already present in state 3. This led us to write the structural analysis of the set of 5 maps as H92 being *"slightly stabilised in a near-mature conformation by interaction with H90"* in state 3, and then *"H92 is further stabilised"* in state 4, where it is clearly resolved. We draw a correspondence between these states and the mtLSU assembly pathway, bearing in mind this is a dynamic process and that some changes in conformation may be gradual and dependent on other events. To address the reviewer's comment, we edited the sentence: *"In state 3, H92 (which contains the target of MRM2 – U3039) becomes slightly stabilised in a near-mature conformation by interaction with H90."*

Reviewer #2

It was a pleasure to read the study by Rebelo-Guioimar et al., who investigated the molecular function and the physiological role of MRM2 in more detail. First, they generated cell lines deficient in the three 2'-O-methyltransferases MRM1, MRM2 and MRM3. Their genome-wide analysis suggests that 2'-O-methylations are only present in the 16S rRNA and that the modifications introduced by MRM2 and MRM3 are interdependent. In the absence of MRM2 mtLSU assembly is stalled at late maturation states accumulating five intermediates, which the authors resolved by cryo-EM. Most interestingly, although mtLSU maturation depends on MRM2, its methyltransferase activity is dispensable suggesting a second role as an assembly factor by remodeling the ribosomal interface side. The study is not only a good complementation to the previous reported mtLSU assembly structures, it also addresses the physiological role of MRM2 and its implications in human mitochondrial diseases.

The experiments performed in this study are well designed and show high quality data. Hence, this manuscript is suitable for publication in Nature Communications. However, I have a few minor issues, which the authors may want to consider:

1) The majority of structural studies follows the ribonucleotides numbering according to their relative position in the mitochondrial genome. The authors may consider adjusting this in their study. (G1145 = G2815; U1369 = U3039; G1370 = G3040)

Following the reviewer's suggestion, we have changed the mt-rRNA numbering.

2) Supplementary Fig. 1: The level of MRM1 decreases in MRM2 KO and MRM2 is reduced in MRM3 KO. Could the authors comment on this? It might be that the reduced 2'-O-methylation of U1369/U3039 in MRM3 KO is caused by the reduced levels of MRM2.

We are grateful for this careful analysis of the knock-out cell data. While we appreciate a slight reduction of MRM1 and MRM2 protein steady-state levels in MRM2 and MRM3 KO cells, respectively, we note that (i) no reduction in the modification status of MRM1 target (G1145/G2815) is observed in MRM2 KO cells and (ii) the very strong reduction in modification of the MRM2 target (U1369/U3039) in MRM3 KO cells is unlikely to be attributable to a slight downregulation of the MRM2 protein. To address the reviewer's comment, we modified the text of the Results section (page 5).

The authors included rescue experiments for MRM2 KO, but similar experiments for MRM1 and MRM3 are missing. This should be included to confirm the specificity of the KO.

The role of MRM2 protein is the focus of the submitted study. The key aim of the MRM2 rescue experiment was to assess whether methylation of U1369/U3039 is required for proper mitoribosome assembly. The two remaining MRM proteins – MRM1 and MRM3 – are subject of further, detailed research in the laboratory. For example, we have performed a rescue experiment for MRM1 and observed a clear recovery of G1145/G2815 modification (measured by RT primer extension), confirming specificity of the MRM1 knock-out (**Fig. R1**). We are currently working on

a manuscript describing the role of MRM1 in mitoribosome assembly (Palenikova *et al.* in preparation), hence we feel that the rescue experiment falls beyond the scope of this study.

CONFIDENTIAL: Figure R1 | Loss of G2815 modification in MRM1 knockout cells (A) Immunoblot analysis showed loss of MRM1 protein in complete (MRM1^{-/-}) and the partial knockout cells (MRM1^{+/-}). Complementation of MRM1^{-/-} with MRM1-FLAG induced for 24 hours with 50 ng/ml doxycycline. TOM22 was used as loading control. **(B)** Schematic representation of primer extension assay. 'Gm' indicates the site of the modification. Dark grey represents the primer binding site. Dark purple indicates primer extension up to the site of the modification. Light purple indicates primer extension up to the site of stalling caused due to the absence of dATP. **(C)** Primer extension assay for HEK293T, MRM1^{+/-}, MRM1^{-/-}, MRM1^{-/-} complemented with MRM1-flag and an IVT control. The intensity of the bands indicates the magnitude of pausing/ stalling. **(D)** The band intensities calculated from the primer extension assays. A ratio of the primer extension stalled at the modification relative to total pausing/stalling events (which includes the pausing at the site of the modification and the fully extended primer terminated due to a lack of dATP). This was normalised relative to the control. Error bars = 1 SEM; n = 4, ns not significant, *p < 0.05, **p < 0.01, unpaired two-tailed Student's t test with Control.

3) Fig. 3b: TOM22 seems to be strongly increased, while the loading (beta-actin) seems to be fine. Is there an increase in mitochondrial mass in MRM2 KO and how can this be explained?

We thank the reviewer for noticing the increase in the steady-state level of TOM22, a subunit of the outer mitochondrial import receptor. This finding was somewhat unexpected, as we show that depletion of MRM2 causes a severe mitochondrial dysfunction phenotype. To confirm this result, we assessed mitochondrial mass by MitoTracker Green staining and subsequent flow cytometric analysis of parental and MRM2 KO cells. We used this compound as its partitioning to mitochondria is independent of the membrane potential of this organelle (which is expected to be affected in MRM2 KO cells due to the observed severe OxPhos impairment). We observed an increased fluorescence intensity measured in MRM2 KO cells when compared to the corresponding parental control. This could be the result of a compensatory mechanism in which cells increase mitochondrial biogenesis as a response to unmet energetic and metabolic demands caused by mitochondrial dysfunction. Other instances of compensatory mechanisms have been reported [Metodiev et al., PLOS Genetics 2014] and we believe this phenomenon requires a deeper understanding. Although we feel it is somewhat out of scope of the present work to pursue this finding further, we included the mitochondrial mass data in the supplement (Supplementary Fig. 2d-e) since it can be useful in the context of other investigations and contributes towards the critical mass to trigger an investigation on compensatory mechanisms in mitochondrial dysfunction. We also added a note on this potential compensatory mechanism in the manuscript (page 6/7).

4) Fig. 4: It would also be great to include a western blot for the density gradient. From the graph in Fig. 4a it is not clear whether 55S monosomes are still formed in the KO. Would fraction 10 correspond to 55S? In addition, the authors could include some assembly factors in their gradient analyses e.g. GTPBP5, which seems to bind downstream of MRM2 to the mtLSU, or other GTPases. Although structurally not resolved, mtLSU intermediates accumulating in MRM2 KO might have other assembly factors bound. Thus, further biochemical analyses might be informative.

The conditions used in our density gradient promote the dissociation of ribosomal subunits by the presence of a chelating agent (EDTA) as well as a reduced concentration of magnesium cations. This was done to analyse mitoribosomal subunits separately, as their presence in monosomes could mask some observations such as the presence of a heterogeneous population of subunits. Given the structural insight, where we show that virtually all mtLSU particles contain the MALSU1:L0R8F8:mtACP anti-association module in the absence of MRM2, if monosomes are formed, their representation should be considerably small.

5) Page 9: This is a strange sentence: "We set out to investigate whether the relevance of MRM2 for mitoribosome biogenesis is due to the chemical modification it introduces in U1369 or to MRM2 acting as a platform for local conformational/compositional changes once bound to assembling mtLSU particles." Please, rephrase.

This sentence has been edited: "Therefore, we asked whether the catalytic activity of MRM2 is essential for mitoribosome biogenesis and/or if the MRM2 protein acts as a

platform for local conformational/compositional changes upon binding to assembling mtLSU particles.”

6) Supplementary Fig. 11: Although this blot is not of good quality, one could still conclude that the overexpression of mitochondrial GTPases affects mitochondrial translation as COX2 is decreased in the parental background if GTPBP5, -7, -8 or -10 are elevated. This is also in agreement with recent studies showing that overexpression of mitochondrial GTPases reduces mitochondrial translation as they act as anti-association factors preventing subunit joining (Maiti et al., 2020; Lavdovskaia et al., 2018). This might also be one of the reasons why their overexpression does not rescue the MRM2 KO phenotype. Maybe worth to include in the discussion.

Having demonstrated that (i) mtLSU accumulates as a near-mature intermediate with disordered domain IV and V (PTC) in the absence of MRM2, (ii) knowing that other assembly factors (some of which with orthologues in mammals) bind the PTC in ribosomes from other systems, and (iii) previous studies [Tan et al., Journal of Bacteriology 2002] showing that overexpression of some small GTPases (ObgE and EngA) rescues the translation deficiency caused by depletion of the corresponding MRM2 orthologue, we set out to investigate whether mammalian mitochondria present this redundancy mechanism. It is not entirely known how the small GTPases are able to rescue ribosome assembly in bacteria. However, since they bind domains IV and V of the LSU rRNA, it is possible to hypothesise that the binding of these factors to assembling ribosomes in which these RNA domains are present in an immature conformation may stabilise their structure and promote proper folding. The same could happen in the case of mtLSU in the absence of MRM2: the large subunit is left with unstructured rRNA (to which MRM2 would be able to bind and stabilise) and which could be folded by, for example, GTPBP5 (ObgE orthologue). There is no known EngA orthologue in human and we decided to also include other GTPBP proteins in this study, some of which are known to participate in mtLSU assembly. This case is relatively different from parental cells, where overexpression of assembly factors could stall the pathway. In the absence of MRM2, the pathway is already stalled, with accumulation of intermediates. Overexpression of assembly factors is used here as a means to favour the interaction of these with the stalled intermediates. In case these act upstream of the accumulated species, additional stalling points could be introduced. However, GTPBP5 and GTPBP10 have been shown to act downstream of MRM2, and thus their overexpression is more likely to contribute to resume assembly if these proteins show redundancy to MRM2 than it is to cause further stalling.

Reviewer #3

This manuscript addresses specific aspects of human mitochondrial ribosome biogenesis. In particular, the authors are studying the involvement of MRM2, a 2'-O methyltransferase active on the mitochondrial large ribosomal subunit RNA, the 16S.

Briefly, the authors start by assigning a single 2'-O methylated residue of 16S to each of MRM1, MRM2, and MRM3.

We note that we assessed 2'-O-methylation in a global analysis of the entire mitochondrial transcriptome. However, we only detected the G1145/G2815, U1369/U3039 and G1370/G3040 sites as being dependent on MRM1, MRM2 and MRM3, therefore, we focused on these sites.

The remainder of the work focusses on MRM2, which is deleted in HEK293T (kidney) cells, and depleted in flies. MRM2 was known to be associated to human MELAS-like syndrome which affects primarily the brain, the nervous system, and the muscles. In human kidney cells, the gene is not essential and delete cells are defective for mitochondrial function, mitochondrial translation,

We use HEK293T cells (a widely used system with adrenal/neural crest ectodermal origin [Graham et al. Journal of General Virology 197; Lin et al. Nat Comm 2014]) as a model to reveal, for the very first time, the molecular, mechanistic role of the *MRM2* gene in the basic biology of mitochondria in mammals. We note that deletion of *MRM2* leads to a very severe mitochondrial dysfunction, including undetectable levels of complex IV. The reviewer mentions that *MRM2* “*is not essential*”, likely referring to the fact that *MRM2* knock-out cells are viable. However, we note that mammalian cells can survive in culture without mtDNA (they are called Rho0 cells) i.e. without any mitochondrial translation, if media are supplemented with pyruvate and uridine (we included both of these supplements in the medium while working with *MRM2* knock-out cells), therefore, the term “essentiality” needs to be reconsidered in the context of mitochondrial dysfunction. Our *in vivo* data clearly show that *MRM2* is essential for organismal viability in a mitochondrial function-dependent manner, hence we had to use weaker drivers for *MRM2* knock-down. We would like to point the reviewer to page 11 of the manuscript, where we state: “*DmMRM2 knock-down was lethal when performed under another ubiquitous driver (act5C-GAL4) or a pan-neuronal driver (nSyb-GAL4), further indicating the essentiality of this protein.*”

and (partially) for large ribosomal subunit assembly.

We note that we do not detect fully matured mtLSU in *MRM2* knock-out cells, therefore, the reference to only partial defect of large ribosomal subunit assembly is incorrect – please see below for further detailed explanation.

In fly, the protein is indispensable for homeostasis, which really questions the pertinence of kidney cells as model (as I understand, this organ is not particularly affected in MELAS-like syndrome).

Please see above. We aimed to establish the mechanistic role for *MRM2* and we consider HEK293T cells suitable for this goal.

The manuscript has been prepared with great care and contains a lot of high-quality high-resolution data.

We appreciate this positive assessment of our work, which is in line with the comment by the other reviewers.

Regretfully, the biological insight is too limited. Essentially, what the authors have shown is that cells deprived of a ribosome assembly factor don't make ribosome well, and consequently loose translation and function.

We are somewhat confused by this comment as our work reveals several novel findings that were clearly greatly appreciated and complimented by the other reviewers as being of “*remarkable quality*”, containing a wealth of experimental data and clearly being “*suitable for publication in Nature Communications*”.

Our work aims at starting to unravel the role of mitochondrial RNA modifications in translation occurring within this organelle. With this in mind, we performed a set of experiments that allowed us to conclude: (i) the known 16S mt-rRNA 2'-O-methyltransferases are specific for their known targets and do not modify other mitochondrial transcripts; (ii) the methylation introduced by MRM2 is dependent on MRM3; (iii) MRM3 acts prior to MRM2 in mtLSU biogenesis; (iii) depletion of MRM2 severely impairs mitochondrial translation, which underpins the observed OxPhos/mitochondrial dysfunction phenotype; (iv) MRM2 participates in the late stages of mtLSU biogenesis as an assembly factor (point acknowledged by the reviewer); (v) the methyltransferase activity of MRM2 is not required for its role in mtLSU biogenesis; (vi) human mitochondria lack redundancy show for the LSU assembly in other organisms. Apart from these molecular, biochemical and structural insights, we also show that, *in vivo*, MRM2 is essential for homeostasis of organisms and its depletion in neurons and muscle cause severe phenotypes. We would like to add that this study was guided by the fact that human patients harbouring mutations in MRM2 have severe clinical presentations (MELAS-like syndrome) for which molecular and mechanistic basis was not known. In addition, throughout this work, several new tools to study diverse aspects of mitochondrial biology were developed/implemented, which could be of use in future work (e.g. qDGMS, heterogeneity analysis of the mitoribosome, *in vivo* model of tissue specific mitochondrial dysfunction indirectly targeting OxPhos, mitoRibo-Seq).

The authors are proposing the MRM2 is part of a quality control mechanism during late stage of mitochondrial ribosome biogenesis, but it is not entirely clear how this works?

In addition to the extensive discussion on the role of MRM2 in the late stage of mitoribosome assembly, which can be found in the manuscript and above, we would like to quote the summary of our finding provided by Reviewer 1: “*MRM2 is essential as an assembly factor that remodels the rRNA during a late stage of the mt-LSU maturation*” and by Reviewer 2 who sums up our study by stating that MRM2 plays a “*role as an assembly factor by remodeling the ribosomal interface side*”. The late-stage rRNA remodelling activity could be considered as a part of quality control. However, to address the reviewer's comment, in the revised manuscript we clearly state how

MRM2 participates in the late stages of mtLSU assembly. To this end, we edited the last paragraph of the Introduction: *We show that MRM2, but not its methyltransferase activity, is essential for mtLSU biogenesis, as it remodels 16S rRNA conformation in the late stages of this process.* Moreover, we changed the last sentence of the abstract to: *This work identifies a key, checkpoint during mtLSU assembly, essential to maintaining mitochondrial homeostasis.*

Indeed MRM2 is not essential in HEK293T cells, implying mitochondrial ribosomes are made in its absence.

This seems to be a repetition and we addressed the issue of essentiality in the context of mitochondrial (dys)function above.

In fact mrm2 -/- cells are producing two types of ribosomes: 1) unmodified mature ribosome engaged in translation and sustaining cell growth, and 2) stalled large ribosomal subunits, which are translationally inactive because they have retained assembly factors masking functional sites (interface). The question is to know why sometimes the assembly process is halted, and why sometimes it goes to fruition. Another important question to answer is what is the respective amounts of mature ribosomes versus stalled precursors produced.

We note that we did not detect mature mtLSU particles in the absence of MRM2 using two following approaches. First, and as presented in the manuscript, the cryoEM classification scheme used in the generation of the reported mtLSU models yielded no mature subunits (Supplementary Fig. 5, Supplementary Fig. 7 and Supplementary Fig. 9a). The state that most resembles a mature mtLSU corresponds to ~1% of the particles and those still contain the MALSU1:L0R8F8:mtACP anti-association module as well as a different conformation of H89, as described in the text (page 7-9 and Discussion). In addition to convincing evidence from thorough classification approaches (Fig. 5, Supplementary Fig. 5, Supplementary Fig. 7 and Supplementary Fig. 9a), particles from state 5 were further subjected to focused classification using a mask containing the anti-association module, but no particles without it were detected (not shown).

It is evident that the very low level of mitochondrial translation occurring in the absence of MRM2 has to arise from mtLSU particles where 16S U1369/U3039 is not 2'-O-methylated (since MRM2 is the only enzyme responsible for this, as also shown in this work, and the successful depletion of this protein), and from which the MALSU1:L0R8F8:mtACP anti-association module has dissociated. However, these events must occur at a frequency below 1%. Please note that these unmodified mtLSU particles have been shown to engage in translation in mitochondria (shown in this work upon complementation of MRM2 KO with catalytically inactive MRM2 mutants, with the lack of the respective RNA modification having been confirmed by RNA MS) as well as in other systems [Tan et al., Journal of Bacteriology 2002].

We also note that the statement “*stalled large ribosomal subunits, which are translationally inactive because they have retained assembly factors masking functional sites (interface)*” is incorrect. If the reviewer is referring to the MALSU1:L0R8F8:mtACP anti-association module, then it interacts with portions of uL14m and bL19m, which do not encompass any of the functional sites of the

mitochondrial ribosome. This module works by being a steric hindrance to the association of assembling mtLSUs with mtSSUs [Brown et al. Nat Struct Mol Biol. 2017].

Also important would be to know by how much production of mature ribosome is reduced (this is particularly important because ribosome profiling illustrates very well that translationally active ribosome translate indifferently from wild-type control ribosomes.)

We clearly state in the manuscript that mature-like mtLSU (state 5) are present at ~1% of total number of particles. Here again, these particles contain the anti-association module. We would like to point the reviewer to page 8: *“With the organisation of H89-93, the configuration progresses to state 5, which presents H68-71 in their mature conformation and is thus the most complete and mature-like state; however, this is also the least populated state (~1% of total number of particles, Supplementary Fig. 7), possibly representing particles that were able to stochastically advance through the pathway without the aid of MRM2.”*

With so many open questions, and the limited biological insights offered, I regret I cannot recommend publication of this manuscript in Nature Communications.

We hope that by providing the above additional information and through addressing the more minor points below, we were able to convince the reviewer regarding the novel findings brought forward with the underlying work and improved the manuscript. These explanation and improvements, together with the enthusiastic assessment by the other reviewers, hopefully make our work suitable for further consideration by the Journal.

Other comments:

-to bring the MRM2 results in perspective, data on MRM1 and MRM3 would have been really very useful (in particular since all three modify the same ribosomal RNA).

We agree with the reviewer that the study of other RNA modifying enzymes is essential to gain an overall understanding of the role of RNA modifications in mitochondrial translation. The roles of MRM1 and MRM3 are studied in our laboratory. However, here we state the rationale behind focusing our attention on MRM2. To produce evidence of equal breadth and deepness, similar/parallel studies would be required which, while not deprecating the findings presented here, are out of the scope of the present study (please see also the response to Reviewer 2).

-Unlike stated in the Discussion, the authors have not established that the catalytic function of MRM2 is not involved in ribosomal subunit assembly (they acknowledge this in the Results section; but somehow they don't in the Discussion). What they have shown, using rescue constructs, which have their own limitations, is that translation of select products resume, which is not the same (especially considering that a yet-to-be-determined level of mature ribosomes are produced in the delete cells...).

We are confused by this comment, since Reviewer 1 was very complimentary regarding this part of the study and pointed out that *the authors reveal that the methyltransferase activity of MRM2 is not essential to the biogenesis of the mt-LSU. Instead, the authors conclude that MRM2 is essential as an assembly factor that remodels the rRNA during a late stage of the mt-LSU maturation. This finding is literally amazing and merits to be highlighted!*

We showed that deletion of MRM2 abolishes mitochondrial translation. Using the proteomics approach, we observed a marked decrease in the steady-state levels of OxPhos subunits. Among the IDs, mt-COX2 was the one presenting the largest decrease in its steady-state amounts between parental and *MRM2* KO; this finding was validated by immunodetection. Furthermore, upon complementation of the *MRM2* KO background with wild-type MRM2, production of mt-COX2 was regained, which strengthened the use of this mtDNA-encoded protein as a proxy for the status of mitochondrial translation.

Complementation of cell lines, as any other technical approaches, have their own limitations. Thinking of that, we made sure to subject integrated cells to a thorough selection and not using unselected cells or single selected clones to mitigate potential associated technical effects. Furthermore, we made sure to validate the lack of catalytic activity of the mutants we studied (Figure 6B) as there was no previous information on this, and the targeted residues were chosen by combining the evaluation of biochemical data from bacterial systems as well as the detailed inspection of the structure of the active site of human MRM2. We verified the ablation of methyltransferase activity by assessing the U3039 modification using RNA MS (Fig. 6b).

Since (i) mtDNA-encoded proteins (including mt-COX2) are translated by mitochondrial ribosomes, (ii) the steady-state level of mt-COX2 is severely decreased in *MRM2* KO (qMS, WB), (iii) expression of mt-COX2 is regained upon complementation of the *MRM2* KO background with wild-type MRM2, and (iv) catalytic variants of MRM2 were validated (RNA MS), absence of mt-COX2 expression in *MRM2* KO cells complemented with two different MRM2 catalytic mutants indicates that these are able to rescue mitochondrial translation.

The remark regarding “*yet-to-be-determined level of mature ribosomes are produced in the delete cells*” is addressed above.

-Why is there so much heterogeneity at the interface on stalled precursor particles in the mutant (at least 5 conformations, if I understand well). Why is it so?

Biological systems are dynamic. As such, it is relatively rare to find rigid entities at the microscale. A good example of this are macromolecular complexes (e.g. ribosomes), which often need to present some flexibility to perform their processes. Structural Biology techniques offer a unique view into this nanoscopic world. However, their methodologies often require that the samples are ordered/immobilised. While not requiring the same extent of orderliness as crystallography, samples need to be frozen in vitreous ice to be analysed by cryoEM. This generates still “pictures” of the biological targets in the state/conformation they were in at the time the sample was frozen. By collecting a large number of these states/conformations, it is possible to generate a

consensus model (which is what is often displayed in scientific works). However, all states/conformations are superimposed in this model, and often these are not deconvolved. Here, endowed with an unusually large number of particles (1.2 million), we set out to investigate further into these states/conformations that are often overlooked. To study heterogeneity (compositional and conformational) in our dataset we used a more classical approach (focused classification) as well as a new, promising and evolving machine learning algorithm (cryoDRNG). The first approach was able to distinguish 5 discrete states in our dataset; the second confirms features seen by the classical approach while also providing a continuous landscape of additional intermediate conformations. All in all, it is unmeaning to consider biological entities (e.g. large multi-component macromolecular complexes) as rigid bodies fixed in a given homogeneous state, conformation or composition.

Are these five, and possibly more conformations, similarly represented in the population of particles? Or are some more abundant than others?

We addressed this in the manuscript (page 7-9) and the relative abundancies are presented in Supplementary Fig. 7, which a part of is presented below.

-Why is it that the MRM2-mediated modification depends upon the MRM3-mediated modification but that the contrary is not true? At least, it's not because the metabolic stability of MRM2 is affected in mrm3 -/- cells (the authors show this but do not comment on it, they could). Is it because cells lacking MRM3 are also defective for ribosome biogenesis, may be at an earlier stage?

Although the targets of MRM2 and MRM3 were known from previous works, it was uncertain whether these methyltransferases would modify other residues in the mitochondrial transcriptome. In this study, in addition to showing the specificity of the mitochondrial methyltransferases MRM1, MRM2 and MRM3, we present evidence of their interdependency. Using a combination of complementary transcriptome-wide

approaches, a reduction in the 2'-O-methylation of U1369/U3039 (introduced by MRM2) and G1370/G3040 was observed upon depletion of MRM3 (which is responsible for modifying G1370/G3040 only). This implies that presence of modified G1370/G3040 (Gm1370/Gm3040) is required for MRM2 to modify the neighbouring residue, U1369/U3039. Given our results, we conclude that MRM3 acts prior to MRM2 on the maturing mtLSU.

A similar point was raised by Reviewer 2 and was addressed by changing the manuscript text on page 5.

REVIEWERS' COMMENTS

Reviewer #1 (Remarks to the Author):

The authors have responded adequately to my comments; I don't have any further questions/comments.

Reviewer #2 (Remarks to the Author):

The authors have addressed the majority of my comments, although I still think that it would be informative to get some insights into the distribution of other assembly factors in the gradient comparing MRM2 KO vs WT (comment 4). Therefore, I would appreciate if the authors at least provide the raw data of the mass spectrometry measurements related to Figure 4. In my opinion raw data of MS analyses should be always provided alongside a manuscript. Apart from that I think that the manuscript is suitable for publication in Nature Communications as mentioned earlier.

Reviewer #3 (Remarks to the Author):

The authors have adequately addressed my comments.

Responses to Reviewers' final comments.

Reviewer #1

The authors have responded adequately to my comments, I don't have any further questions/comments.

We are grateful for the positive assessment of our work

Reviewer #2

The authors have addressed the majority of my comments, although I still think that it would be informative to get some insights into the distribution of other assembly factors in the gradient comparing MRM2 KO vs WT (comment 4). Therefore, I would appreciate if the authors at least provide the raw data of the mass spectrometry measurements related to Figure 4. In my opinion raw data of MS analyses should be always provided alongside a manuscript. Apart from that I think that the manuscript is suitable for publication in Nature Communications as mentioned earlier.

We are grateful for the positive assessment of our work. We agree with the Reviewer and provided the raw data related to Figure 4 as Supplementary Data 2.

Reviewer #3

The authors have adequately addressed my comments.

We are grateful for this supportive assessment of our submission.

Other changes in the manuscript:

We changed Fig.1 as one of the values of the "RT stop score" was not statistically significant upon careful reanalysis. This does not impact on the conclusions drawn from Fig 1a, as the "cleavage protection score" is highly statistically significant for the U3039 site.

We also corrected Fig.7c – upon reanalysis of the raw data we realised that the mt:ND1 molecular weight and western blot were incorrect. Here again, this change does not impact the conclusions of Fig. 7b as similar pattern of developmental OXPHOS changes is observed.